# LGGBench: a Holistic Benchmark for Large Graph Generation

### Abstract

The escalating demand for robust graph data sharing between organizations has propelled the development of methodologies that assess the efficacy and privacy of these shared graphs. We present LGGBench, a comprehensive benchmark designed to evaluate large graph generation methods across multiple dimensions crucial to proprietary data sharing. This benchmark integrates a diverse array of large graph datasets, sophisticated graph generation techniques, and comprehensive evaluation schemes to address the current shortcomings in graph data sharing. Our benchmark evaluates the generated graphs in terms of fidelity, utility, privacy, and scalability. Fidelity is assessed through graph statistical metrics, while utility measures the practical applicability of synthetic graphs in real-world tasks. Privacy is ensured through robust mechanisms against various inference attacks, and scalability is demonstrated through the benchmark's ability to handle extensive graph datasets efficiently. Through extensive experiments, we compare existing graph generation methods, highlighting their strengths and limitations across different types of graphs and evaluation metrics. The benchmark provides a holistic approach to evaluate and improve graph generation techniques, facilitating safer and more effective data sharing practices. Code is available[1].

## 1 Introduction

The rapid growth of graph data in recent years has necessitated advanced methods for extracting insights from complex datasets. Graph Neural Networks (GNNs) (Kipf & Welling, 2016; Hamilton et al., 2017; Zhou et al., 2020; Gasteiger et al., 2018) have emerged as a powerful tool to address this need, leveraging their ability to model and learn from the intricate relationships inherent in graph-structured data. Applications of GNNs span a wide array of domains, including recommendation systems on social networks (Fan et al., 2019; Zhang et al., 2022a), anomaly detection in transaction networks (Lin et al., 2024), link prediction on citation graphs (Cho, 2024), and more (Guo et al., 2021; Gasteiger et al., 2021; Dong et al., 2023).

To fuel modern graph analytical models and enhance collaborative research, companies are increasingly interested in sharing large-scale graph data with external researchers (Huang et al., 2022; Darabi et al., 2022; Bauer et al., 2024; Li et al., 2023b). However, due to privacy concerns, companies are turning to graph generation techniques as a viable solution instead of direct data sharing, as illustrated in Figure 1. Notable techniques in this field include GraphRNN (You et al., 2018), GraphVAE (Simonovsky & Komodakis, 2018), and GraphEBM (Liu et al., 2021), which facilitate the synthesis of graphs that can be shared with researchers. While generating realistic graphs provides substantial benefits, two critical aspects must be addressed: (1) the generated graphs should adhere to the original graphs' distribution to facilitate meaningful research, and (2) the privacy of the proprietary data should be protected. This poses a significant challenge: *how can we comprehensively evaluate the shared graphs to ensure effective graph analysis while protecting sensitive proprietary data?*

**Prior works.** To assess the quality of these graph generation approaches, prior evaluation schemes have been proposed to measure different aspects, but they fall short of providing a comprehensive assessment that covers the key elements for large graph generation, as illustrated in Table 1. GraphGT (Du et al., 2021)

---

[1]https://anonymous.4open.science/r/LGGBench-93F7

Figure 1: Large graph generation techniques are employed to create a synthetic version of the proprietary target graph, allowing public researchers to work with large-scale graph data while avoiding direct data leakage.

emphasizes fidelity in graph generation but does not support attributed or labeled graphs, nor does it assess scalability, utility, or privacy aspects. On the other hand, the General Graph Generators benchmark (Xiang et al., 2022) broadens the evaluation by incorporating scalability but still lacks support for attributed or labeled graphs, as well as utility and privacy evaluations. A holistic benchmark is needed that supports attributed and labeled graphs and assesses all key aspects including scalability, fidelity, utility, and privacy.

**Present work.** In response to the challenges in existing benchmarks, we introduce a **L**arge **G**raph **G**eneration **Bench**mark, named **LGGBench**, to provide a holistic assessment tool for large graph generation methods. Our benchmark represents a significant advancement over previous efforts by providing a comprehensive assessment tool tailored for large-scale graph generation methods. First, our benchmark supports the evaluation of complex graph types such as attributed and labeled graphs, which are commonly adopted in real-world applications but have not been sufficiently addressed in prior benchmarks. Furthermore, LGGBench fills the existing gaps by offering a more robust and multifaceted evaluation framework that directly addresses the challenge of synthetic graph assessment in real-world graph sharing scenarios. To empower the development of advanced graph generation methods, we release our benchmark and provide a publicly accessible website.

Our benchmark employs the following comprehensive metrics: (1) *Fidelity* is quantified through the distance of graph statistical metrics between original graphs and synthetic graphs. (2) *Utility* is measured by comparing model performance on synthetic graphs with that on the original graphs for specific tasks. (3) *Scalability* is demonstrated by the time and space consumption of graph generation methods. (4) *Privacy* is assessed through susceptibility to inference attacks, where less successful attacks denote better privacy protection. The integration of these evaluation aspects facilitates a holistic evaluation for practical and secure graph generation in proprietary graph sharing applications.

We summarize our contributions as follows:

- **Taxonomy for Graph Generation Approaches.** We present a new taxonomy that categorizes graph generation methods based on their support for various graph types, including attributed and labeled graphs. This classification highlights the limitations of existing approaches and provides a clear pathway for extending current techniques to handle complex, real-world graph properties.

- **Multi-Metric Evaluation Framework for Graph Sharing.** We introduce a holistic evaluation framework that assesses multiple critical metrics—fidelity, utility, scalability, and pri-

Table 1: Existing graph generation benchmarks and their evaluation aspects. (Attr.&Label: supports attributed and labeled graphs. Scale.: measures scalability. Fidel.: measures fidelity. Util.: measures utility. Priv.: evaluates privacy level.)

| Name | Attribute & Label | Scalability | Fidelity | Utility | Privacy |
|---|---|---|---|---|---|
| GraphGT (Du et al., 2021) | ✗ | ✗ | ✓ | ✗ | ✗ |
| General Gen. (Xiang et al., 2022) | ✗ | ✓ | ✓ | ✗ | ✗ |
| LGGBench (Ours) | ✓ | ✓ | ✓ | ✓ | ✓ |

Table 2: Comparison of graph generation methods. Methods shown in **bold** are those evaluated in our experiments, with proper modifications applied if a specific functionality was not supported (✗).

| Category | Subcategory | Method | SingleInG | SingleOutG | GenStru | GenAttr | GenLabel |
|---|---|---|---|---|---|---|---|
| Probabilistic | Traditional | **SBM** Holland et al. (1983) | ✓ | ✓ | ✓ | ✗ | ✗ |
| | | **R-MAT** Chakrabarti et al. (2004) | ✓ | ✓ | ✓ | ✗ | ✗ |
| | | **SW** Watts & Strogatz (1998) | ✓ | ✓ | ✓ | ✗ | ✗ |
| | | **BA** Barabási & Albert (1999) | ✓ | ✓ | ✓ | ✗ | ✗ |
| | | Kronecker Leskovec et al. (2010) | ✓ | ✓ | ✓ | ✗ | ✗ |
| | | MAG Kim & Leskovec (2012) | ✓ | ✓ | ✓ | ✗ | ✗ |
| | | TrillionG Park & Kim (2017) | ✓ | ✓ | ✓ | ✗ | ✗ |
| | RNN | GraphRNN You et al. (2018) | ✗ | ✗ | ✓ | ✗ | ✗ |
| | | DeepGMG Li et al. (2018) | ✗ | ✗ | ✓ | ✗ | ✗ |
| | Autoencoder | GraphVAE Simonovsky & Komodakis (2018) | ✗ | ✗ | ✓ | ✓ | ✓ |
| | Transformer | **CGT** Yoon et al. (2023) | ✓ | ✗ | ✓ | ✓ | ✓ |
| | Diffusion | **GraphMaker** Li et al. (2023a) | ✓ | ✓ | ✓ | ✓ | ✓ |
| | | ILE Bergmeister et al. (2023) | ✗ | ✗ | ✓ | ✗ | ✗ |
| Utility-oriented | GradMatch | **GCond** Jin et al. (2021) | ✓ | ✓ | ✓ | ✓ | ✓ |
| | DistriMatch | **SimGC** Xiao et al. (2024) | ✓ | ✓ | ✓ | ✓ | ✓ |
| Privacy-oriented | K-Anonymity | **K-Anonimity** Yoon et al. (2023) | ✓ | ✓ | ✓ | ✓ | ✓ |
| | DP | **NodeEdgeDP** Dwork (2006) | ✓ | ✓ | ✓ | ✓ | ✓ |
| | | PrivGraph Yuan et al. (2023) | ✓ | ✓ | ✓ | ✓ | ✓ |

vacy—specifically tailored for graph sharing tasks. In addition, our framework incorporates sensitivity tests that systematically analyze the robustness of each metric under varying conditions, thereby providing a comprehensive and reliable assessment of graph sharing methods.

- **Extensive Experimental Validation.** Through extensive experiments, we compare various graph generation methods, highlighting their strengths and limitations across diverse graph types and evaluation metrics, and provide directions for future advancements.

## 2 Related Work

Large-scale graph sharing has become an increasingly important area of research, motivated by the need for collaborative work between businesses and the research community. Following the success of synthetic data generation (Wang et al., 2023; Shen et al., 2024), the literature contains a wide range of graph generation methodologies that exhibit varying characteristics and are evaluated on multiple aspects.

### 2.1 Graph Generation Approaches

Graph generation methods have evolved considerably over the years, driven by the need to accurately replicate the complex structures found in real-world networks while also addressing specific application requirements such as utility and privacy. In Table 2, we group these methods into three primary categories based

on their generation objectives: probabilistic, utility-oriented, and privacy-oriented approaches. We further distinguish subcategories by the models used in these approaches.

**Probabilistic Methods.** Early work in graph generation predominantly leveraged probabilistic models. Traditional models such as the Stochastic Block Model (SBM) Holland et al. (1983), R-MAT Chakrabarti et al. (2004), Small-World (SW) Watts & Strogatz (1998), and BA Barabási & Albert (1999) focus on reproducing the inherent structural characteristics of networks. These models emphasize properties such as community structure, scale-free degree distributions, and short path lengths. More scalable variants, including Kronecker Leskovec et al. (2010), MAG Kim & Leskovec (2012), and TrillionG Park & Kim (2017), extend these ideas to very large graphs while primarily targeting structural replication.

In recent years, the integration of deep generative models has expanded the capabilities of probabilistic approaches. Pioneering graph generation studies often involve the generation of a set of small graphs, including GraphRNN You et al. (2018) and GraphVAE Simonovsky & Komodakis (2018). Autoencoder-based methods like GraphVAE Simonovsky & Komodakis (2018) and transformer-based methods such as CGT Yoon et al. (2023) have demonstrated the potential to jointly generate structural features along with node attributes and labels. Diffusion-based models, including GraphMaker Li et al. (2023a) and ILE Bergmeister et al. (2023), further improve the generation process by gradually transforming noise into structured graphs, thereby achieving higher fidelity in graph synthesis.

**Utility-Oriented Methods.** Utility-oriented approaches focus on generating graphs that retain high analytical value for downstream tasks such as node classification, clustering, and link prediction. These methods are often designed to closely match the statistical properties and distributions observed in the original data. Techniques under this category, predominantly adopted in graph condensation approaches such as GCond Jin et al. (2021) and SimGC Xiao et al. (2024), employ strategies like gradient matching and distribution alignment. By doing so, they ensure that the generated graphs are not only structurally realistic but also maintain the attribute and label information critical for preserving the utility of the data.

**Privacy-Oriented Methods.** As concerns over data privacy have grown, there has been a surge in graph generation techniques that incorporate privacy guarantees. Privacy-oriented methods are designed to produce synthetic graphs that safeguard sensitive information while still capturing essential graph characteristics. For example, approaches based on K-Anonymity ensure that individual node identities are obfuscated by having at least $K$ duplicates. In addition, differential privacy Dwork (2006) has been integrated into graph synthesis, which provides rigorous privacy guarantees by carefully balancing the trade-off between data utility and privacy protection.

This categorization underscores the diversity of approaches in graph generation. While probabilistic methods lay the foundational principles for structural replication, utility-oriented and privacy-oriented methods extend these foundations to meet application-specific demands. Our experiments focus on evaluating a selection of these methods, which are highlighted in bold in Table 2.

While the reviewed methods have significantly advanced the field of graph generation, we note that many lack certain functionalities critical for modern graph sharing scenarios. In particular, several methods were not originally designed to simultaneously generate attributed graphs with labels, which limits their applicability in contexts where the complete semantic richness of the graph is essential. To address this issue, our work adapts these methods, augmenting their capabilities to support the full spectrum of functionalities—including support for single input graphs and single output graphs, structural generation, as well as attribute and label synthesis—required for practical large-scale graph sharing.

## 2.2 Existing Benchmarks

Synthetic graph evaluation in existing graph generation benchmarks involves assessing key aspects for large graph generation:

- **Fidelity** ensures synthetic graphs accurately represent original ones through various graph- and node-level statistics Gupta et al. (2022); You et al. (2018); Du et al. (2021); Darabi et al. (2022).

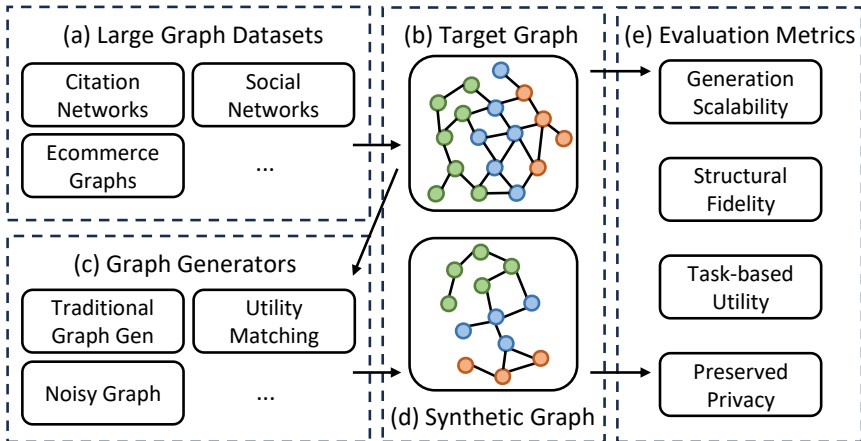

Figure 2: An overview of LGGBench.

- **Utility** is tested by comparing model performance on synthetic and original graphs to ensure consistency Yoon et al. (2023); Palowitch et al. (2022); Jin et al. (2021).

- **Scalability** refers to a graph generator's time and space consumption when handling large original and synthetic graphs Darabi et al. (2022).

- **Privacy** evaluation estimates risks from various inference attacks Wu et al. (2021); Zhang et al. (2022b); Gong & Liu (2018); He et al. (2021b).

Existing graph generation benchmarks facilitate graph comparison in a few aspects but fall short in providing comprehensive assessments that cover all key aspects of graph generation, as shown in Table 1. Our benchmark aims to provide a holistic evaluation framework supporting diverse graph types, ensuring high scalability, fidelity, utility, and privacy.

## 3 Benchmarking Large Graph Generation

Figure 2 depicts the workflow of our benchmark consisting of data collection, graph generation, and graph evaluation. The scope of this benchmark is described in Appendix A. We start by defining the task of large graph generation to facilitate proprietary graph data sharing in real-world scenarios. Then, we introduce the evaluation metrics in multiple aspects to measure the performance of the graph generation methods.

### 3.1 Problem Statement

Graph data sharing boosts the development of graph generation techniques and hence necessitates an effective benchmark to holistically evaluate large synthetic graphs. Specifically, we focus on a graph generation mechanism $g(\cdot)$ that produces a $N'$-node synthetic graph $\mathcal{S} = \{\mathbf{A}', \mathbf{X}', \mathbf{Y}'\}$ from a $N$-node target graph $\mathcal{T} = \{\mathbf{A}, \mathbf{X}, \mathbf{Y}\}$, where $\mathbf{A}$ is the adjacency matrix, $\mathbf{X}$ denotes the node features, and $\mathbf{Y}$ represents the node labels. An effective graph generation method ensures that $\mathcal{S}$ preserves crucial characteristics of $\mathcal{T}$ without compromising the proprietary data. Concretely, we introduce a comprehensive evaluation scheme spanning fidelity, utility, scalability, and privacy, applicable to both homogeneous and heterogeneous large graphs. The evaluation schemes of these aspects are given in the following sections.

### 3.2 Fidelity Evaluation

This evaluation aspect pivots around the question of *how similar does a synthetic graph appear compared to the target graph?* To answer this, we adopt a series of graph statistical metrics to estimate the distance

between the target graph and the synthetic graph. We detail the fidelity metrics in Appendix B and introduce a subset of these metrics below. We start by introducing the graph-level and node-level statistics.

**Graph-level Statistics.** We list the key statistics (Gupta et al., 2022; Bojchevski et al., 2018) as follows. For the $i$-th graph-level statistics, we compute $m_i(G) \in \mathbb{R}$ to quantitatively describe graph $G$.

*Mean Degree:* the average number of edges connected to a vertex. $\text{MeanDeg}(G) = \frac{2|E|}{|V|}$ , where $|E|$ is the number of edges and $|V|$ is the number of vertices.

*Power Law Exponent:* measures the exponent $\alpha$ in a power law distribution $P(k) \sim k^{-\alpha}$, where $P(k)$ is the proportion of nodes in graph $G$ with degree $k$. The exponent is estimated by fitting a power law model to the node degree distribution. We denote $\text{PLE}(G) = \alpha$.

**Node-level Statistics** We adopt the commonly used node-level statistics (You et al., 2018; Gupta et al., 2022) below. For the $i$-th statistical metric, we compute $M_i(G) \in \mathbb{R}^{|G|}$ to describe the graph using a node distribution.

*Degree*: The degree of a node $v$ is the number of edges connected to $v$. $\deg(v) = |\{u \mid (u, v) \in E\}|$

*Closeness Centrality*: The closeness centrality of a node $v$ is the reciprocal of the sum of the shortest path distances from $v$ to all other nodes. $\text{CC}(v) = \frac{1}{\sum_{u \neq v} d(v,u)}$ where $d(v, u)$ is the shortest path distance between nodes $v$ and $u$.

**Graph Distances.** Based on the computed graph-level and node-level statistics, we further calculate the following distance metrics between a pair of graphs.

*Deviation of graph-level statistics:* for the $i$-th statistics, we compute $\mathcal{D}_i = \text{abs}\left(m_i\left(G'\right) - m_i\left(G\right)\right)$.

In summary, by computing the distance metrics, we provide a quantitative estimation of how similar a synthetic graph appears in its structure and features compared to the target graph.

### 3.3 Utility Evaluation

This evaluation targets the question: *does the synthetic graph contain the training information as the target graph?* To address this, we define the following task-specific evaluation schemes to assess the utility of the synthetic graph.

**Model Transfer.** This evaluation scheme involves training an optimal model on a synthetic graph, then evaluating this model back on the target graph (Jin et al., 2021). A highly utilizable synthetic graph should yield an optimal model that also work well on the target graph. This process is described by:

$$
\begin{aligned}
&\mathcal{L}(\text{GNN}_{\boldsymbol{\theta}_{\mathcal{S}}}(\mathbf{X}, \mathbf{A}), \mathbf{Y}) \\
\text{s.t.} \quad &\boldsymbol{\theta}_{\mathcal{S}} = \arg\min_{\boldsymbol{\theta}} \mathcal{L}(\text{GNN}_{\boldsymbol{\theta}}(\mathbf{X}', \mathbf{A}'), \mathbf{Y}')
\end{aligned}
\tag{1}
$$

where $\boldsymbol{\theta}$ denotes the learnable parameters of a GNN model, $\boldsymbol{\theta}_{\mathcal{S}}$ represents the optimal GNN parameters learned on the synthetic graph, $\mathcal{L}$ is a loss function evaluating the node classification performance.

In addition to node classification, evaluating link prediction performance is crucial to assess the generalizability of the learned representations. Link prediction involves estimating the likelihood that an edge exists between a pair of nodes, which can reveal important structural properties of the graph. Common evaluation metrics for link prediction include the Area Under the ROC Curve (AUC) and Average Precision (AP).

Similarly, to quantify link prediction performance, one can define a loss function $\mathcal{L}_{\text{lp}}$ that measures the discrepancy between the predicted link probabilities and the actual connectivity. This evaluation can be formulated as:

$$
\begin{aligned}
&\mathcal{L}_{\text{lp}}(\text{GNN}_{\boldsymbol{\theta}_{\mathcal{S}}}(\mathbf{X}, \mathbf{A}), \mathbf{A}^*) \\
\text{s.t.} \quad &\boldsymbol{\theta}_{\mathcal{S}} = \arg\min_{\boldsymbol{\theta}} \mathcal{L}_{\text{lp}}(\text{GNN}_{\boldsymbol{\theta}}(\mathbf{X}', \mathbf{A}'), \mathbf{A}'^*)
\end{aligned}
\tag{2}
$$

Table 3: An overview of datasets.

| Dataset | Nodes | Train/Val/Test Nodes | Edges | Features | Classes |
|---|---|---|---|---|---|
| Amazon-Photos (Shchur et al., 2018) | 7,650 | 4,590/1,530/1,530 | 119,081 | 745 | 8 |
| WikiCS (Mernyei & Cangea, 2020) | 11,701 | 580/1,769/5,847 | 215,603 | 300 | 10 |
| Amazon-Computers (Shchur et al., 2018) | 13,752 | 8,251/2,750/2,751 | 245,861 | 767 | 10 |
| Amazon-Ratings (Platonov et al., 2023) | 24,492 | 12,246/6,123/6,123 | 93,050 | 300 | 5 |
| Flickr (Zeng et al., 2019) | 89,250 | 44,625/22,312/22,313 | 449,878 | 500 | 7 |
| Ogbn-Arxiv (Hu et al., 2020) | 169,343 | 90,941/29,799/48,603 | 1,157,799 | 128 | 40 |
| Reddit (Hamilton et al., 2017) | 232,965 | 153,431/23,831/55,703 | 57,307,946 | 602 | 41 |

where $\mathbf{A}^*$ denotes the ground truth binary adjacency matrix of the target graph, indicating the presence or absence of edges, and $\mathcal{L}_{\mathrm{lp}}$ is typically instantiated as a binary cross-entropy loss or a similar objective suited for link prediction tasks.

During evaluation, high performance on both node classification and link prediction tasks indicates that the synthetic graph effectively captures the underlying structural and relational properties of the target graph, thereby supporting robust model transfer.

**Performance Correlation.** We evaluate multiple models on both target graph and synthetic graphs individually. Let the performance scores of these models on the target graph be $P_{\mathcal{T}} = \{p_{G1}, p_{G2}, \ldots, p_{Gn}\}$ and on the synthetic graph be $P_{\mathcal{S}} = \{p_{G'1}, p_{G'2}, \ldots, p_{G'n}\}$. We then calculate their performance correlations using metrics including the Pearson and Spearman correlation coefficients following literature (Li et al., 2023a; Yoon et al., 2023). To provide a comprehensive evaluation on the model performance, we select a wide range of GNNs including GCN (Kipf & Welling, 2016), GraphSAGE (Hamilton et al., 2017), SGC (Wu et al., 2019), GIN (Xu et al., 2018), APPNP (Gasteiger et al., 2018), GAT (Veličković et al., 2018), GATv2 (Brody et al., 2021), and GCNII (Chen et al., 2020), to train on the involved graphs.

### 3.4 Scalability Evaluation

The evaluation on scalability focuses on the question: *how scalable is the graph generator?* To address this, we assess graph generation approaches based on their time and space efficiency when handling large-scale graphs. These metrics ensure that the graph generator is practical for real-world applications where large graph datasets are common.

**Memory consumption.** We assess the graph generation methods in Table 6 with large synthetic graph size and measure their consumed memory, depicting the memory efficiency of graph generation approaches. It is notable that graph generation involves generating an adjacency matrix with up to $N'^2$ elements, often posing a scalability challenge when the generation size is large. These memory-consuming approaches include GCond, SimGC, CGT, and GraphMaker.

**Time consumption.** We evaluate the generation time with large synthetic graph size, illustrating the time efficiency of the generation methods. Due to the large number of nodes and edges in both target graphs and synthetic graphs, it is important to design scalable generation methods when handling large-scale graphs.

### 3.5 Privacy Evaluation

This evaluation aims at measuring the privacy level of the synthetic graph with the help of training-based attacks (Wu et al., 2021).

**Threat Model.** the attacker has full $G'$, white-box GNN $f_{\theta'}(\cdot)$ trained on $G'$, part of $G = \{A, X, Y\}$, and interested in learning knowledge of $G$. We further assume the attacker has a subgraph $G^{sub}$ with controlled nodes $V^{attack}$ and target nodes $V^{target}$, inducing a masked $X^{sub}$ and a masked $A^{sub}$.

**Inference Attacks.** We employ the commonly used inference attacks to evaluate how well the synthetic graph preserves private information. (1) *Graph property inference attack Zhang et al. (2022b).* The attacker

tries to infer sensitive graph-level properties of $G$. (2) *Node attribute inference attack Hsieh & Li (2021); Gong & Liu (2018); Wu et al. (2021).* The attacker tries to reconstruct $X^{sub}$. (3) *Edge membership inference attack He et al. (2021b).* The attacker tries to reconstruct $A^{sub}$. (4) *Node membership inference attack Olatunji et al. (2021); He et al. (2021a).* given a node $v$ and its $L$-hop neighborhood, the attacker tries to determine if $v \in V$ from the posterior of $f_{\theta'}(\cdot)$.

## 4 Experiments

To validate the effectiveness of our benchmark, we introduce the experimental settings that encompass real-world graphs exhibiting diverse characteristics and various graph generation mechanisms. We further present empirical results that highlight the strengths and limitations of existing graph generation methods, offering insights for advancing techniques applicable to proprietary graph sharing scenarios.

### 4.1 Experimental Settings

To demonstrate the effectiveness and comprehensiveness of our proposed benchmark, we adopt common large-scale graph datasets and prevalent graph generation methods. We detail the datasets, graph generation methods, hyperparameter settings, and unified graph postprocessing in Appendix C, D, and E, respectively.

**Runtime Settings.** For each valid combination of graph dataset and graph generation approach, we follow the default hyper-parameter settings provided by the respective methods to generate a set of synthetic graphs with different sizes. For training-based evaluations, we repeat all experiments 5 times on one V100-32GB GPU and report the average performance with standard deviation.

**Graph Postprocessing** For methods that generate unrealistic dense adjacency matrices, we adopt a sampling-based postprocessing step that results in an undirected and unweighted graph. For example, GCond (Jin et al., 2021) inherently produces a dense graph structure, so we perform postprocessing on the generated graph to produce a sparse graph structure.

We specify the synthetic graph's degree distribution using a power-law distribution $P(k) \sim k^{-\alpha}$ given its prevalence in real-world graphs (Leskovec et al., 2005; Faloutsos et al., 1999), where $P(k)$ denotes the proportion of nodes in the graph with degree $k$. By default, we fit the original graph's node degrees using a power-law estimator[2] so that the synthetic graph shares a similar degree distribution with the original graph. Finally, we reallocate the synthetic node degrees using the power-law distribution and resample neighbors for each synthetic node.

### 4.2 Fidelity Analysis

We evaluate the fidelity of the synthetic graphs with respect to the original Ogbn-Arxiv graph by comparing a broad set of graph-level statistics, as shown in Table 4. For each graph generation method, we produce an equal-sized synthetic graph ($N' = N$) in a best-effort manner, and then measure how closely its properties match those of the target graph.

- **Traditional Methods Maintain Overall Structural Fidelity.** Traditional graph generators such as SBM, RMAT, SW, and BA exhibit superior structural fidelity, as reflected by their lower mean rankings across multiple metrics. Notably, the differentially private (DP) method achieves a competitive ranking when the privacy budget is sufficiently loose (e.g., $\epsilon = 20$), thereby limiting the amount of modification required. In contrast, learning-based approaches (e.g., GraphMaker, GCond, and SimGC) not only encounter scalability issues when generating graphs of comparable size but also demonstrate large deviations in other metrics, indicating the challenge of generating large, realistic graphs.

- **Insufficient Preservation of Fine-grained Graph Characteristics.** The results further reveal significant variations in fine-grained metrics, such as global eigenvector centrality (GloEC) and

---

[2]https://pypi.org/project/powerlaw/

Table 4: Fidelity evaluation on Ogbn-Arxiv. The distances between synthetic and original graph metrics and are highlighted in gray. The Rank column presents the mean ranking over the metrics.

| Method | #Nodes | #Edges | Deg | #CC | LCC | Dia | #Tri | PLE | EEnt | Trans | GloEC | GloCC | Ho% | Rank |
|---|---|---|---|---|---|---|---|---|---|---|---|---|---|---|
| SBM | 169,343 | 1,280,076 | 15.1 | 44 | 169,300 | 11 | 5.4e+03 | 1.3906 | 0.9883 | 0.0007 | 0.0019 | 0.0006 | 68.2 | - |
| RMAT | 169,343 | 2,315,359 | 27.3 | 82 | 169,262 | 7 | 1.2e+04 | 1.3319 | 0.9788 | 0.0004 | 0.0019 | 0.0004 | 11.6 | - |
| SW | 169,343 | 1,185,398 | 14.0 | 1 | 169,343 | 10 | 2.6e+06 | 3.2808 | 0.9997 | 0.5010 | 0.0024 | 0.5075 | 8.0 | - |
| BA | 169,343 | 1,185,373 | 14.0 | 1 | 169,343 | 6 | 1.4e+04 | 3.0957 | 0.9720 | 0.0007 | 0.0013 | 0.0010 | 11.3 | - |
| CGT ($\epsilon$=10) | 169,343 | 163,880 | 1.9 | 5,463 | 31 | 4 | 0.0e+00 | 3.9333 | 0.9693 | 0.0000 | 0.0021 | 0.0000 | 100.0 | - |
| GraphMaker | 29,799 | 32,192 | 2.2 | 13,015 | 14,821 | 23 | 3.2e+02 | 2.2144 | 0.9063 | 0.0040 | 0.0022 | 0.0008 | 59.1 | - |
| GCond | 2,000 | 15,161 | 15.2 | 568 | 1,433 | 8 | 5.1e+01 | 1.5395 | 0.8323 | 0.0001 | 0.0077 | 0.0034 | 2.0 | - |
| SimGC | 1,727 | 269 | 0.3 | 1,526 | 161 | 5 | 9.4e+01 | 4.3777 | 0.5431 | 0.0245 | 0.0057 | 0.0084 | 40.5 | - |
| KAnon (k=100) | 169,343 | 2,366,146 | 27.9 | 1 | 169,343 | 5 | 3.5e+04 | 1.9169 | 0.9961 | 0.0015 | 0.0023 | 0.0016 | 20.8 | - |
| DP ($\epsilon$=20) | 169,343 | 2,460,593 | 29.1 | 1 | 169,343 | 6 | 2.3e+06 | 1.4782 | 0.9738 | 0.0145 | 0.0010 | 0.0333 | 34.9 | - |
| SBM | 0 | 122,277 | 1.4 | 43 | 43 | 14 | 2.2e+06 | 0.1517 | 0.0723 | 0.0155 | 0.0014 | 0.2255 | **2.8** | 3.6 |
| RMAT | 0 | 1,157,560 | 13.6 | 81 | 81 | 18 | 2.2e+06 | 0.2104 | 0.0628 | 0.0158 | 0.0014 | 0.2257 | 53.8 | 5.5 |
| SW | 0 | 27,599 | 0.3 | **0** | **0** | 15 | 4.0e+05 | 1.7385 | 0.0837 | 0.4848 | 0.0019 | 0.2814 | 57.4 | 4.9 |
| BA | 0 | **27,574** | **0.3** | **0** | **0** | 19 | 2.2e+06 | 1.5534 | 0.0560 | 0.0155 | 0.0008 | 0.2251 | 54.1 | 3.4 |
| CGT ($\epsilon$=10) | 0 | 993,919 | 11.8 | 5,462 | 169,312 | 21 | 2.2e+06 | 2.3910 | 0.0533 | 0.0162 | 0.0016 | 0.2261 | 34.6 | 6.3 |
| GraphMaker | 139,544 | 1,125,607 | 11.5 | 13,014 | 154,522 | **2** | 2.2e+06 | 0.6721 | **0.0097** | 0.0122 | 0.0017 | 0.2253 | 6.3 | 4.8 |
| GCond | 167,343 | 1,142,638 | 1.5 | 567 | 167,910 | 17 | 2.2e+06 | **0.0028** | 0.0837 | 0.0161 | 0.0072 | 0.2227 | 63.4 | 6.2 |
| SimGC | 167,616 | 1,157,530 | 13.4 | 1,525 | 169,182 | 20 | 2.2e+06 | 2.8354 | 0.3729 | 0.0083 | 0.0052 | 0.2177 | 24.9 | 6.8 |
| KAnon (k=100) | 0 | 1,208,347 | 14.3 | **0** | **0** | 20 | 2.2e+06 | 0.3746 | 0.0801 | 0.0147 | 0.0018 | 0.2246 | 44.6 | 5 |
| DP ($\epsilon$=20) | 0 | 1,302,794 | 15.4 | **0** | **0** | 19 | **1.0e+05** | 0.0641 | 0.0578 | **0.0017** | **0.0005** | **0.1928** | 30.5 | **3.3** |
| Target | 169,343 | 1,157,799 | 13.7 | 1 | 169,343 | 25 | 2.2e+06 | 1.5423 | 0.9160 | 0.0162 | 0.0005 | 0.2261 | 65.4 | - |

global closeness centrality (GloCC). This divergence suggests that existing methods fall short in reproducing nuanced statistical properties.

- **Lack of Structure-Label Connectivity Preservation.** Our analysis also indicates that current graph generation approaches do not adequately preserve structure-label connectivity patterns. This is evident from the discrepancies in the homophily metric (Ho%), where the synthetic graphs fail to maintain the label-dependent connectivity patterns.

Overall, these observations highlight the strengths of traditional generators in preserving overall structural fidelity, while also exposing their limitations in maintaining fine-grained and label-dependent characteristics. The results motivate further research into novel graph synthesis techniques that can more accurately mirror the complex properties of large-scale real-world graphs.

### 4.3 Utility Analysis

Table 5 presents the utility metrics for various graph generation methods on the Ogbn-Arxiv dataset. The table reports the accuracies obtained by different graph models (MLP, SGC, GCN, GAT, Cheb, SAGE, and APPNP) on synthetic graphs, alongside the Mean Accuracy. Additionally, Pearson and Spearman correlations are calculated over the accuracies of an array of graph models trained separately on the synthetic and target graphs. The target row, obtained by training and evaluating the same model on the original graph, serves as the reference for the accuracies of different models.

- **Utility Preservation of Privacy-Focused Methods.** Privacy-preserving methods, such as K-Anonymity ($k = 100$) and DP ($\epsilon = 20$), demonstrate robust utility preservation when privacy constraints are moderately relaxed. In these cases, the synthetic graphs exhibit performance metrics that closely mirror those of the target graph. For instance, the DP method achieves a Mean Accuracy of 67.22% in the top block, which is comparable to the target accuracy of 67.96%.

- **Competitive Accuracy of Utility-Focused Methods.** Utility-focused methods, notably GCond and SimGC, achieve competitive accuracy compared to traditional graph generation techniques. Their superior performance is largely attributed to their task-oriented training objectives, which emphasize gradient matching between the synthetic and target graphs. This tailored approach ensures that key task-specific patterns are preserved, as reflected by higher Pearson and Spearman

Table 5: Utility evaluation on Ogbn-Arxiv. Pearson and Spearman correlations are calculated on the accuracies across an array of graph models separately trained on two graphs. A referential target accuracy is given by training and evaluating the same model on the target graph.

| Method | MLP | SGC | GCN | GAT | Cheb | SAGE | APPNP | MeanAcc | Pear. (↑) | Spear. (↑) | Rank |
|---|---|---|---|---|---|---|---|---|---|---|---|
| SBM | $43.64_{\pm0.23}$ | $60.20_{\pm0.58}$ | $58.40_{\pm0.52}$ | $59.95_{\pm0.65}$ | $52.48_{\pm0.49}$ | $54.48_{\pm1.48}$ | $55.57_{\pm0.65}$ | $54.96_{\pm0.32}$ | - | - | - |
| RMAT | $46.21_{\pm0.22}$ | $61.98_{\pm0.04}$ | $60.18_{\pm0.18}$ | $58.37_{\pm0.97}$ | $50.27_{\pm1.26}$ | $50.10_{\pm0.74}$ | $57.71_{\pm0.03}$ | $54.98_{\pm0.05}$ | - | - | - |
| SW | $46.44_{\pm0.65}$ | $63.77_{\pm0.04}$ | $63.60_{\pm0.14}$ | $60.85_{\pm0.16}$ | $50.26_{\pm0.90}$ | $53.41_{\pm0.17}$ | $59.06_{\pm0.10}$ | $56.77_{\pm0.06}$ | - | - | - |
| BA | $46.23_{\pm0.30}$ | $63.05_{\pm0.06}$ | $59.56_{\pm0.13}$ | $57.48_{\pm0.79}$ | $51.55_{\pm0.92}$ | $51.23_{\pm0.64}$ | $57.76_{\pm0.19}$ | $55.27_{\pm0.09}$ | - | - | - |
| CGT ($\epsilon$=10) | $46.48_{\pm0.11}$ | $60.81_{\pm0.35}$ | $61.97_{\pm0.50}$ | $56.95_{\pm2.28}$ | $56.83_{\pm0.18}$ | $60.97_{\pm0.22}$ | $62.60_{\pm0.19}$ | $58.09_{\pm0.38}$ | - | - | - |
| GraphMaker | $46.29_{\pm0.09}$ | $60.82_{\pm0.46}$ | $58.15_{\pm0.87}$ | $59.30_{\pm0.81}$ | $54.29_{\pm0.20}$ | $56.39_{\pm0.37}$ | $55.29_{\pm0.01}$ | $55.79_{\pm0.15}$ | - | - | - |
| GCond | $44.87_{\pm0.10}$ | $61.33_{\pm0.32}$ | $62.41_{\pm0.13}$ | $58.94_{\pm0.34}$ | $55.16_{\pm0.21}$ | $55.54_{\pm0.15}$ | $64.11_{\pm0.41}$ | $57.48_{\pm0.02}$ | - | - | - |
| SimGC | $44.93_{\pm0.16}$ | $61.99_{\pm0.09}$ | $62.39_{\pm0.16}$ | $60.39_{\pm1.16}$ | $55.65_{\pm1.42}$ | $59.08_{\pm0.09}$ | $62.21_{\pm0.36}$ | $58.09_{\pm0.14}$ | - | - | - |
| KAnon (k=100) | $54.29_{\pm0.04}$ | $67.55_{\pm0.04}$ | $67.83_{\pm0.03}$ | $67.58_{\pm0.10}$ | $63.37_{\pm0.63}$ | $62.71_{\pm0.14}$ | $65.88_{\pm0.25}$ | $64.17_{\pm0.13}$ | - | - | - |
| DP ($\epsilon$=20) | $60.68_{\pm0.02}$ | $67.91_{\pm0.02}$ | $70.47_{\pm0.04}$ | $71.28_{\pm0.05}$ | $67.30_{\pm0.53}$ | $71.07_{\pm0.05}$ | $61.83_{\pm0.06}$ | $67.22_{\pm0.07}$ | - | - | - |
| SBM | $11.90_{\pm0.23}$ | $8.64_{\pm0.58}$ | $12.80_{\pm0.52}$ | $11.75_{\pm0.65}$ | $14.97_{\pm0.49}$ | $16.66_{\pm1.48}$ | $14.27_{\pm0.65}$ | $13.00_{\pm0.32}$ | $0.89_{\pm0.04}$ | $0.65_{\pm0.07}$ | 8.2 |
| RMAT | $9.33_{\pm0.22}$ | $6.86_{\pm0.04}$ | $11.02_{\pm0.18}$ | $13.33_{\pm0.97}$ | $17.18_{\pm1.26}$ | $21.04_{\pm0.74}$ | $12.13_{\pm0.03}$ | $12.98_{\pm0.05}$ | $0.65_{\pm0.01}$ | $0.33_{\pm0.10}$ | 6.6 |
| SW | $9.10_{\pm0.65}$ | $5.07_{\pm0.04}$ | $7.60_{\pm0.14}$ | $10.85_{\pm0.16}$ | $17.19_{\pm0.90}$ | $17.73_{\pm0.17}$ | $10.78_{\pm0.10}$ | $11.19_{\pm0.06}$ | $0.72_{\pm0.05}$ | $0.52_{\pm0.04}$ | 5.2 |
| BA | $9.31_{\pm0.30}$ | $5.79_{\pm0.06}$ | $11.64_{\pm0.13}$ | $14.22_{\pm0.79}$ | $15.90_{\pm0.92}$ | $19.91_{\pm0.64}$ | $12.08_{\pm0.19}$ | $12.69_{\pm0.09}$ | $0.67_{\pm0.01}$ | $0.33_{\pm0.10}$ | 6.2 |
| CGT ($\epsilon$=10) | $9.06_{\pm0.11}$ | $8.03_{\pm0.35}$ | $9.23_{\pm0.50}$ | $14.75_{\pm2.28}$ | $10.62_{\pm0.18}$ | $10.17_{\pm0.22}$ | $7.24_{\pm0.19}$ | $9.87_{\pm0.38}$ | $0.90_{\pm0.06}$ | $0.42_{\pm0.09}$ | 5.1 |
| GraphMaker | $9.25_{\pm0.09}$ | $8.02_{\pm0.46}$ | $13.05_{\pm0.87}$ | $12.40_{\pm0.81}$ | $13.16_{\pm0.20}$ | $14.75_{\pm0.37}$ | $14.55_{\pm0.01}$ | $12.17_{\pm0.15}$ | $0.89_{\pm0.02}$ | $0.61_{\pm0.00}$ | 7.1 |
| GCond | $10.67_{\pm0.10}$ | $7.51_{\pm0.32}$ | $8.79_{\pm0.13}$ | $12.76_{\pm0.34}$ | $12.29_{\pm0.21}$ | $15.60_{\pm0.15}$ | $5.73_{\pm0.41}$ | $10.48_{\pm0.02}$ | $0.85_{\pm0.02}$ | $0.46_{\pm0.05}$ | 5.5 |
| SimGC | $10.61_{\pm0.16}$ | $6.85_{\pm0.09}$ | $8.81_{\pm0.16}$ | $11.31_{\pm1.16}$ | $11.80_{\pm1.42}$ | $12.06_{\pm0.09}$ | $7.63_{\pm0.36}$ | $9.87_{\pm0.14}$ | $0.94_{\pm0.01}$ | $0.42_{\pm0.02}$ | 5.0 |
| KAnon (k=100) | $\mathbf{1.25_{\pm0.04}}$ | $1.29_{\pm0.04}$ | $3.37_{\pm0.03}$ | $4.12_{\pm0.10}$ | $4.08_{\pm0.63}$ | $8.43_{\pm0.14}$ | $\mathbf{3.96_{\pm0.25}}$ | $3.78_{\pm0.13}$ | $\mathbf{0.90_{\pm0.00}}$ | $0.58_{\pm0.07}$ | 2.9 |
| DP ($\epsilon$=20) | $5.14_{\pm0.02}$ | $\mathbf{0.93_{\pm0.02}}$ | $\mathbf{0.73_{\pm0.04}}$ | $\mathbf{0.42_{\pm0.05}}$ | $\mathbf{0.15_{\pm0.53}}$ | $\mathbf{0.07_{\pm0.05}}$ | $8.01_{\pm0.06}$ | $\mathbf{0.74_{\pm0.07}}$ | $0.74_{\pm0.01}$ | $\mathbf{0.86_{\pm0.00}}$ | **2.7** |
| Target | $55.54_{\pm0.00}$ | $68.84_{\pm0.00}$ | $71.20_{\pm0.00}$ | $71.70_{\pm0.00}$ | $67.45_{\pm0.00}$ | $71.14_{\pm0.00}$ | $69.84_{\pm0.00}$ | $67.96_{\pm0.00}$ | 1.00 | 1.00 | - |

Table 6: Scalability evaluation – memory usage (RAM), GPU memory usage (VRAM), and total running time of different graph generation methods.

| Method | RAM (GB) | VRAM (GB) | Time (s) | Rank |
|---|---|---|---|---|
| SBM | 1.4 | **0** | 104 | 3.0 |
| RMAT | **0.1** | **0** | 127 | 2.7 |
| SW | 1.2 | **0** | 11 | **2.0** |
| BA | 1.3 | **0** | 10 | **2.0** |
| CGT ($\epsilon$=10) | 2.5 | 19.0 | 3,623 | 8.0 |
| GraphMaker | 10.7 | 31.9 | 15,211 | 9.7 |
| GCond | 5.0 | 31.9 | 3,157 | 8.7 |
| SimGC | 2.0 | 31.8 | 383 | 7.3 |
| KAnon (k=100) | 1.8 | **0** | 105 | 4.0 |
| DP ($\epsilon$=20) | 1.6 | **0** | **9** | 2.3 |

correlation values and more favorable ranking metrics. In contrast, methods such as SBM and GraphMaker, which primarily focus on replicating the overall graph distribution, tend to overlook task-specific utility.

Overall, the presented utility analysis underscores the importance of aligning graph generation objectives with the specific requirements of the target tasks. While traditional methods may excel in matching statistical distributions, privacy-perturbation and utility-focused approaches provide improved performance, ensuring that utility is maintained for practical applications.

### 4.4 Scalability Analysis

We assess the scalability of various graph generation methods and examine the trade-offs between computational efficiency and privacy guarantees. Specifically, we measure system RAM, GPU VRAM, and total runtime for generating the synthetic graphs. Additionally, we investigate how varying privacy parameters affect resource utilization and the structural properties (*e.g.*, edge count) of the generated graphs.

Table 7: Impact of privacy parameters ($k$ in K-Anonymity, $\epsilon$ in NodeEdgeDP) on memory usage (RAM), total running time, and number of generated edges. A smaller privacy budget $\epsilon$ in DP or a larger value of $k$ in K-Anonymity both indicate better privacy protection.

| Privacy Param | RAM (GB) | Time (s) | #Edges |
|---|---|---|---|
| $\epsilon$=20 | 1.6 | 9 | 2.5e+06 |
| $\epsilon$=10 | 35.6 | 112 | 1.9e+08 |
| $\epsilon$=5 | 378.3 | 1461 | 2.0e+09 |
| $\epsilon$=1 | 945.8 | 1696 | 1.1e+10 |
| k=5 | 9.6 | 128 | 2.2e+06 |
| k=100 | 1.8 | 109 | 2.4e+06 |
| k=500 | 1.5 | 122 | 2.4e+06 |
| k=1000 | 1.5 | 114 | 2.4e+06 |

We conduct experiments on a single NVIDIA V100-32GB GPU. For each method, we record the RAM and VRAM usage in gigabytes (GB), as well as the overall time in seconds. Table 6 summarizes the performance metrics for different graph generation methods, and Table 7 reports the impact of privacy parameters on resource consumption and graph scale.

**Scalability of Graph Generation Methods.** As shown in Table 6, traditional graph generation techniques such as SBM, RMAT, SW, and BA demonstrate excellent scalability. These methods use minimal RAM and zero VRAM, achieving very fast generation times. In contrast, learning-based approaches (e.g., CGT, GraphMaker, GCond, SimGC) require significant GPU memory, primarily due to the need to store large adjacency matrices and gradients during training. This overhead is reflected in their increased execution times and higher resource consumption, indicating current limitations in scaling these methods to very large graphs.

**Impact of Privacy Parameters.** Table 7 reveals a marked impact of privacy parameters on both resource consumption and graph structure. For the differential privacy (DP) approach, a stricter privacy budget (i.e., lower $\epsilon$) leads to exponential increases in resource demands and graph density. For instance, lowering $\epsilon$ from 20 to 1 results in a dramatic rise in RAM usage (from 1.6GB to nearly 946GB) and runtime (from 9s to 1696s), alongside an increase in the number of generated edges from 2.5e+06 to 1.1e+10. This trend underscores the substantial computational cost of strong privacy guarantees in DP-based graph generation.

Conversely, the K-Anonymity method exhibits a more stable performance profile. Although a smaller $k$ value (e.g., $k = 5$) incurs higher RAM usage (9.6GB) and slightly longer execution times, increasing $k$ beyond a threshold (from 100 to 1000) has negligible impact on both memory and runtime. Moreover, the number of edges remains almost constant at approximately $2.4 \times 10^6$. This indicates that once a certain level of anonymity is reached, further increases in $k$ do not incur additional computational penalties.

The findings highlight a critical trade-off in graph generation between computational efficiency and privacy strength. While traditional and K-Anonymity methods scale well with minimal resource overhead, the evaluated DP method faces significant scalability challenges as privacy constraints tighten. These results suggest a need for further research into resource-efficient algorithms that deliver robust privacy guarantees without compromising scalability.

### 4.5 Privacy Analysis

In this section, we evaluate the privacy protection capabilities of various synthetic graph generation methods by assessing their resistance to attribute and edge membership inference attacks. Table 8 summarizes the experimental results where higher Attribute Mean Squared Error (Attr MSE) and lower Edge Area Under the Curve (Edge AUC) indicate better privacy protection. For reference, the *Self Attack* row reports the performance when the attack model is both trained and evaluated on the target graph, thus representing the best achievable attack performance.

Table 8: Privacy evaluation. An unsuccessful attack yields high MSE in attribute inference and low AUC in edge inference, indicating a higher privacy level of the synthetic graph.

| Method | Attr MSE ($\uparrow$) | Edge AUC ($\downarrow$) | Rank |
|---|---|---|---|
| SBM | $0.9362_{\pm 0.0011}$ | $0.5029_{\pm 0.0004}$ | 6.5 |
| RMAT | $0.9958_{\pm 0.0003}$ | $0.5058_{\pm 0.0009}$ | 4.5 |
| SW | $0.9849_{\pm 0.0001}$ | $0.5346_{\pm 0.0023}$ | 6.5 |
| BA | $0.9958_{\pm 0.0001}$ | $0.5645_{\pm 0.0019}$ | 6.5 |
| CGT ($\epsilon=10$) | $0.9350_{\pm 0.0008}$ | $0.5309_{\pm 0.0021}$ | 8.0 |
| GraphMaker | $0.9544_{\pm 0.0012}$ | $0.5024_{\pm 0.0018}$ | 5.5 |
| GCond | $0.9590_{\pm 0.0004}$ | $0.4958_{\pm 0.0053}$ | 4.5 |
| SimGC | $1.0141_{\pm 0.0059}$ | $\mathbf{0.4593_{\pm 0.0508}}$ | **2.0** |
| KAnon (k=100) | $\mathbf{1.3971_{\pm 0.0022}}$ | $0.5542_{\pm 0.0016}$ | 4.5 |
| DP ($\epsilon=20$) | $1.1765_{\pm 0.0119}$ | $0.6323_{\pm 0.0016}$ | 6.0 |
| Self Attack | $0.8705_{\pm 0.0006}$ | $0.7794_{\pm 0.0017}$ | - |

**Privacy Attacks.** The primary goal is to quantify how well each synthetic graph preserves privacy by measuring the effectiveness of inference attacks:

- **Attribute Mean Squared Error (Attr MSE)** reflects the error of an attribute inference attack model trained on the synthetic graph and evaluated on the target graph. A higher MSE implies that the synthetic graph does not reveal detailed attribute information, thus ensuring better privacy.

- **Edge Area Under the Curve (Edge AUC)** measures the performance of an edge membership inference attack. Lower AUC values indicate that the synthetic graph obscures edges effectively, enhancing edge-level privacy.

**K-Anonymity and DP's Tendency to Protect Attribute-Level Privacy.** We observe that the privacy-preserving methods tend to yield notably high Attr MSE values. This suggests that these methods are effective at protecting attribute-level information. However, these gains in attribute privacy do not always translate to similar improvements in edge privacy, as indicated by their relatively higher Edge AUC values compared to other methods. This also leads to an overall moderate privacy level for these methods.

**Privacy Preservation of Utility-Focused Methods.** Notably, the evaluated utility-focused methods, such as GCond and SimGC, exhibit strong privacy protection. In particular, SimGC achieves the lowest Edge AUC, implying that its approach of condensing high-level training information while discarding individual-specific details may simultaneously enhance privacy. This observation suggests that a well-balanced design can simultaneously deliver both high utility and robust privacy.

## 5 Multi-Aspect Analysis and Visualization

To jointly examine performance across multiple aspects, we provide additional discussion in Appendix F. Moreover, comprehensive visualizations of both real and synthetic graphs are presented in Appendix G.

## 6 Conclusions

The introduction of LGGBench marks a significant advancement in the evaluation of large-scale graph generation methods. By expanding on the limited evaluation aspects of previous benchmarks, LGGBench offers a comprehensive assessment tool that supports various graph types. It evaluates key metrics, including fidelity, utility, scalability, and privacy, facilitating robust analysis of synthetic graph quality. This holistic benchmark not only ensures effective and secure graph sharing but also promotes the development of advanced graph generation techniques for collaborative research.

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

## A  Scope of LGGBench

In this section, we delineate the boundaries and focus of our LGGBench benchmark to ensure it is both targeted and comprehensive in evaluating large graph generation methods.

**Scope of Graph Datasets.** Our benchmark is primarily designed for homogeneous graphs that are both attributed and single-labeled. This focus is motivated by the prevalence of such graph structures in real-world business applications, where the interplay between node attributes and singular labels is critical for downstream tasks. By concentrating on this category, LGGBench can provide detailed and relevant insights into the performance of graph generation methods under conditions that closely mirror practical scenarios.

**Scope of Graph Generation Methods.** LGGBench is tailored to evaluate graph generation techniques that are capable of synthesizing graphs with the same number of nodes as the original graph. This restriction ensures a one-to-one correspondence between the synthetic and proprietary graphs, thereby allowing for a more accurate comparison of key graph properties. In doing so, the benchmark favors methods that maintain structural consistency and capture the inherent statistical distributions of the original graphs, which is crucial for both fidelity and downstream utility.

**Scope of Evaluation Aspects.** Our evaluation framework focuses on the aspects that are most pertinent to secure and effective graph sharing scenarios, including fidelity, utility, scalability, and privacy. These aspects are crucial for practical reasons. Fidelity guarantees that the synthetic graphs accurately mirror the statistical properties of the original graphs, ensuring meaningful analysis. Utility confirms that downstream tasks can be effectively performed on the generated data. Scalability is essential to handle large-scale graphs efficiently, and privacy is critical to protect sensitive information while enabling collaborative research. Since it is impractical to exhaustively perform all the evaluation metrics within these aspects, we select metrics that can be efficiently measured on large-scale synthetic graphs in the experiments.

**Extending Outside the Scope.** While LGGBench is designed with a specific focus, it is inherently modular and extensible. Future work can broaden the benchmark to encompass additional graph types and evaluation criteria. For example, the utility of heterogeneous graphs—those with multiple node types and relations—can be assessed using models such as Heterogeneous Graph Neural Networks (HGNNs). This flexibility allows LGGBench to evolve alongside advancements in graph analytics and generation techniques, ensuring its continued relevance in diverse research and application settings.

## B  Fidelity Metrics

**Graph-level Statistics.** We list the key statistics (Gupta et al., 2022; Bojchevski et al., 2018) as follows. For the $i$-th graph-level statistics, we compute $m_i(G) \in \mathbb{R}$ to quantitatively describe graph $G$.

*Mean Degree:* the average number of edges connected to a vertex. $\text{MeanDeg}(G) = \frac{2|E|}{|V|}$ , where $|E|$ is the number of edges and $|V|$ is the number of vertices.

*Power Law Exponent:* measures the exponent $\alpha$ in a power law distribution $P(k) \sim k^{-\alpha}$, where $P(k)$ is the proportion of nodes in graph $G$ with degree $k$. The exponent is estimated by fitting a power law model to the node degree distribution. We denote $\text{PLE}(G) = \alpha$.

*Relative Edge Distribution Entropy:* a measure of the heterogeneity of the edge distribution across vertices. $\text{EdgeEnt}(G) = -\sum_k \frac{P(k)}{\log(P(k))}$, where $P(k)$ is the proportion of edges that connect to vertices of degree $k$.

*Largest Connected Component Size:* the size of the largest connected component in the graph, expressed as the number of vertices in that component. $\text{LCC}(G) = \max_{C \in \mathcal{C}} |V(C)|$, where $\mathcal{C}$ is the set of all connected components in the graph.

*Number of Components:* the total number of connected components in the graph. $\#\text{CC}(G) = |\mathcal{C}|$.

*Wedge Count:* the total number of length-two paths. $\#\text{Wedge}(G) = \sum_{v \in V} \binom{k_v}{2}$, where $k_v$ is the degree of vertex $v$.

*Triangle Count:* the number of triangles in the graph, where a triangle is a set of three vertices all connected by edges. $T = \frac{1}{6} \sum_{v \in V} \binom{k_v}{2}$.

*Global Clustering Coefficient:* measures the overall level of clustering in the graph. $\text{CF} = \frac{3T}{\sum_{v \in V} \binom{k_v}{2}}$.

*Mean Betweenness Centrality:* the average betweenness centrality across all vertices, where betweenness centrality of a vertex $v$ is the proportion of all shortest paths that pass through $v$. $\text{BC}(G) = \frac{1}{|V(G)|} \sum_{v \in V(G)} \text{BC}(v)$.

*Mean Closeness Centrality:* the average closeness centrality, which is the reciprocal of the average shortest path length from a vertex to all other vertices. $\text{CC}(G) = \frac{1}{|V(G)|} \sum_{v \in V(G)} \frac{1}{\sum_{u \in V(G), u \neq v} d(v,u)}$.

*Edge homophily ratio (Zhu et al., 2020):* the ratio of the number of edges connecting nodes of the same class to the total number of edges. $\text{Homo} = |\{(u,v) \in E \mid c(u) = c(v)\}|/|E|$, where $c(v)$ represents the label of node $v$.

**Node-level Statistics** We adopt the commonly used node-level statistics (You et al., 2018; Gupta et al., 2022) below. For the $i$-th statistical metric, we compute $M_i(G) \in \mathbb{R}^{|G|}$ to describe the graph using a node distribution.

*Degree*: The degree of a node $v$ is the number of edges connected to $v$. $\deg(v) = |\{u \mid (u,v) \in E\}|$

*Closeness Centrality*: The closeness centrality of a node $v$ is the reciprocal of the sum of the shortest path distances from $v$ to all other nodes. $\text{CC}(v) = \frac{1}{\sum_{u \neq v} d(v,u)}$ where $d(v,u)$ is the shortest path distance between nodes $v$ and $u$.

*Betweenness Centrality*: The betweenness centrality of a node $v$ is the sum of the fraction of all-pairs shortest paths that pass through $v$. $\text{BC}(v) = \sum_{s \neq v \neq t} \frac{\sigma_{st}(v)}{\sigma_{st}}$ where $\sigma_{st}$ is the total number of shortest paths from node $s$ to node $t$, and $\sigma_{st}(v)$ is the number of those paths that pass through $v$.

*Clustering Coefficient*: The clustering coefficient of a node $v$ measures the degree to which nodes in a graph tend to cluster together. It is defined as the ratio of the number of triangles connected to $v$ over the number of possible triangles. $\text{CF}(v) = \frac{2T(v)}{\deg(v)(\deg(v)-1)}$ where $T(v)$ is the number of triangles through node $v$.

**Graph Distances.** Based on the computed graph-level and node-level statistics, we further calculate the following distance metrics between a pair of graphs.

*Deviation of graph-level statistics:* for the $i$-th statistics, we compute $\mathcal{D}_i = \text{abs}\,(m_i\,(G') - m_i\,(G))$.

*Maximum Mean Discrepancy (MMD) of graph statistics or features You et al. (2018):* let $\mathbb{M} = \{M_1, ..., M_k\}$, where each $Mi(G)$ is a univariate distribution, such as degree distribution or clustering coefficient distribution. We compute the MMD distance between the two distributions as a distance metric between the target and synthetic graphs.

## C   Large-scale Graph Datasets

We use widely employed node classification datasets from various domains and provide their statistics in Table 3. We follow the public splits available in these datasets. If public splits are not provided, we perform a random split of 60% training nodes, 20% validation nodes, and 20% test nodes.

- *Citation Networks.* This network type is collected by capturing the connections between academic entities. Ogbn-Arxiv (Zeng et al., 2019) is part of the Open Graph Benchmark (Hu et al., 2020) and consists of a citation network of computer science papers from arXiv, where the task is to predict the subject area of each paper. The WikiCS dataset is a graph-based dataset representing article categories and cross-references within Wikipedia, specifically focused on computer science topics.

- *Social Networks.* This network type is constructed by representing users as nodes in the graph. Reddit (Zeng et al., 2019) is a social network dataset where nodes represent posts, edges represent user interactions, and the labels indicate the subreddit to which each post belongs.

- *Web Service Graphs.* This graph type involves entities such as photos, movies, *etc.*, acting as the nodes in the graph. Flickr (Zeng et al., 2019) is a photo-sharing network where nodes represent images, edges represent shared properties between images, and labels correspond to tags. Amazon-Computers and Amazon-Photo (Shchur et al., 2018) are co-purchase networks, where nodes represent products in different categories.

We also present the graph-level statistics in Table 9 for fidelity evaluation. These include the number of nodes and edges, mean degree (Deg), number of connected components (#CC) and the size of the largest connected component (LCC), diameter (Dia), number of triangles (#Tri), power-law exponent (PLE) of the degree distribution, relative edge distribution entropy (EEnt), graph transitivity (Trans), global eigenvector centrality (GloEC), global closeness centrality (GloCC), and edge homophily ratio (Ho%).

Table 9: Graph-level statistics of target graphs.

| Dataset | #Nodes | #Edges | Deg | #CC | LCC | Dia | #Tri | PLE | EEnt | Trans | GloEC | GloCC | Ho% |
|---|---|---|---|---|---|---|---|---|---|---|---|---|---|
| Photos | 7,650 | 119,081 | 31.1 | 136 | 7,487 | 10 | $7.2 \times 10^5$ | 1.3421 | 0.9403 | 0.1773 | 0.0047 | 0.4040 | 82.7 |
| Wikics | 11,701 | 215,603 | 36.9 | 356 | 11,311 | 10 | $3.2 \times 10^6$ | 1.3750 | 0.9099 | 0.2623 | 0.0047 | 0.4527 | 65.4 |
| Computers | 13,752 | 245,861 | 35.8 | 314 | 13,381 | 10 | $1.5 \times 10^6$ | 1.3349 | 0.9309 | 0.1077 | 0.0036 | 0.3441 | 77.7 |
| Ratings | 24,492 | 93,050 | 7.6 | 1 | 24,492 | 46 | $1.1 \times 10^5$ | 4.4617 | 0.9828 | 0.3163 | 0.0024 | 0.5816 | 38.0 |
| Flickr | 89,250 | 449,878 | 10.1 | 1 | 89,250 | 8 | $6.4 \times 10^4$ | 1.5013 | 0.9470 | 0.0039 | 0.0012 | 0.0330 | 31.9 |
| Arxiv | 169,343 | 1,157,799 | 13.7 | 1 | 169,343 | 25 | $2.2 \times 10^6$ | 1.5423 | 0.9160 | 0.0162 | 0.0005 | 0.2261 | 65.4 |
| Reddit | 232,965 | 57,307,946 | 492.0 | 1 | 232,965 | 8 | $8.4 \times 10^9$ | 1.1915 | 0.9395 | 0.2443 | 0.0009 | 0.5793 | 75.6 |

## D Graph Generation Methods

We consider the following approaches:

- *Traditional Structural Generation.* A major category of graph generation studies involves generating only structures, including SBM Holland et al. (1983), R-MAT Chakrabarti et al. (2004), Small World (SW) Watts & Strogatz (1998), and BA Barabási & Albert (1999). Due to the lack of generated attributes and labels, we assign node features and labels to the generated structures using degree matching and Gaussian-distributed features. Concretely, we sort the nodes according to their degree ranking and match them with labels based on the corresponding ranking in the original graph. Then, for each class, we estimate the parameters of a Gaussian distribution from the original node features and generate synthetic features by sampling from these class-specific distributions.

- *Learning-based Scalable Graph Generation.* We adopt GraphMaker and CGT for comparison. GraphMaker (Li et al., 2023a) is a diffusion model-based approach for graph generation, learning to denoise a graph in order to reconstruct its structure and feature distribution. CGT (Yoon et al., 2023) generates multiple small computation graphs to preserve the training utility of the synthetic dataset.

- *Utility Matching.* These methods generate graphs from their original versions while preserving training information. We adopt GCond (Jin et al., 2021) and SimGC (Xiao et al., 2024) for this category.

- *Privacy-preserving.* We adopt two lines of approaches: K-Anonymity (Samarati & Sweeney, 1998) and Differential Privacy (Dwork, 2006). The adopted K-Anonymity method Yoon et al. (2023) enforces that each node feature and edge appears at least $K$ times in the graph, thereby reducing the risk of re-identification. In contrast, NodeEdgeDP achieves privacy by adding calibrated noise to node features and subsequently perturbing the adjacency matrix based on a defined privacy budget, ensuring differential privacy while maintaining graph utility.

# E   Hyper-parameter Settings

By default, we generate synthetic graphs of equal size to the original graphs ($N' = N$). If generation fails due to Out-of-Memory (OOM) on a single V100-32GB GPU or Out-of-Time (OOT) over 12 hours, we generate the largest possible graph that fits into the VRAM or can be completed within a reasonable time, respectively. We then detail the hyperparameter settings for the graph generation methods.

- *SBM, R-MAT, SW, BA.* We provide implementations in our benchmark, including synthetic attribute generation. All hyper-parameters are inferred from the corresponding original graphs to generate graphs with realistic structures.

- *GCond.* We follow the implementation[3]. For GCond, we choose a learning rate of 0.01, use 2 GNN layers, and employ SGC as the backbone GNN. For herding, we set the reduction rate according to the target generation size.

- *SimGC.* We follow the implementation[4] and set the number of GNN layers to 2, the number of hidden units to 256, the number of condensation loops to 1500, the learning rate for adjacency matrix learning to 0.01, and 0.05 for feature learning.

- *CGT.* We follow the official implementation[5] to set the hyper-parameters. We choose the number of noise edges to be 2, a learning rate of 0.001, and a hidden dimension of 64.

- *GraphMaker.* We follow the implementation[6] and choose a hidden size of 64, 2 GNN layers, and 3 diffusion steps.

- *K-Anonymity.* We follow prior work Yoon et al. (2023) for enforcing $k$-anonymity in graph data, ensuring that every node is indistinguishable from at least $k - 1$ others. In our experiments, we set the default $k = 100$.

- *NodeEdgeDP.* We follow the definitions of Differential Privacy Dwork (2006), applying DP to both nodes and edges. We set the default privacy budget $\epsilon = 10.0$ for both node- and edge-level DP, to balance privacy, utility, and scalability. For example, a smaller privacy budget results in numerous noisy edges, significantly increasing processing time.

# F   Multi-Aspect Analysis

We summarize the method rankings in individual aspects in Figure 3a-3d and provide a cross-aspect ranking in Figure 3e.

**Privacy Protection vs. Other Aspects.** Although a high level of privacy may coexist with other aspects, such as the high utility achieved in K-Anonymity (k=100), it is impractical for current methods to fulfill all aspects simultaneously. For example, DP ($\epsilon$=20) achieves high utility and fidelity while being relatively scalable; however, it sacrifices privacy due to its relaxed privacy constraints. Producing high-quality graphs that consider all these metrics remains an open problem.

**Performance Gap of Learning-based Methods.** Notably, most learning-based methods achieve relatively poor performance in the multifaceted evaluation. The main challenge lies in generating large, effective graph structures. We encourage future research on learning-based graph generators to be applicable for real-world, large-scale graphs.

# G   Visualization

We present visualization of original graph structures in Figure 4, synthetic graph structures in Figure 5, and synthetic graph structures produced by different methods in Figure 6.

---

[3] https://github.com/ChandlerBang/GCond
[4] https://github.com/BangHonor/SimGC
[5] https://github.com/minjiyoon/CGT
[6] https://github.com/Graph-COM/GraphMaker

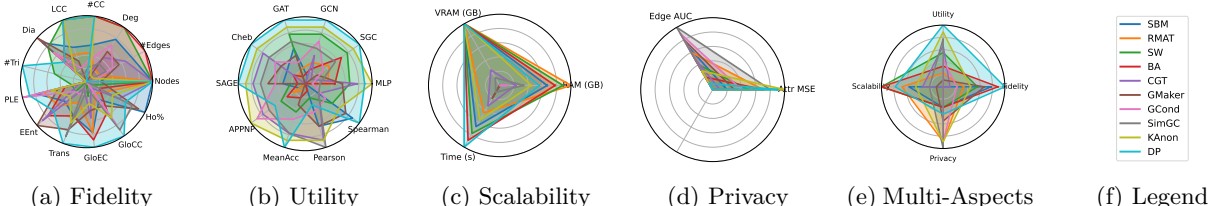

Figure 3: Method rankings in each aspect (a-d) and across aspects (e) on Ogbn-Arxiv.

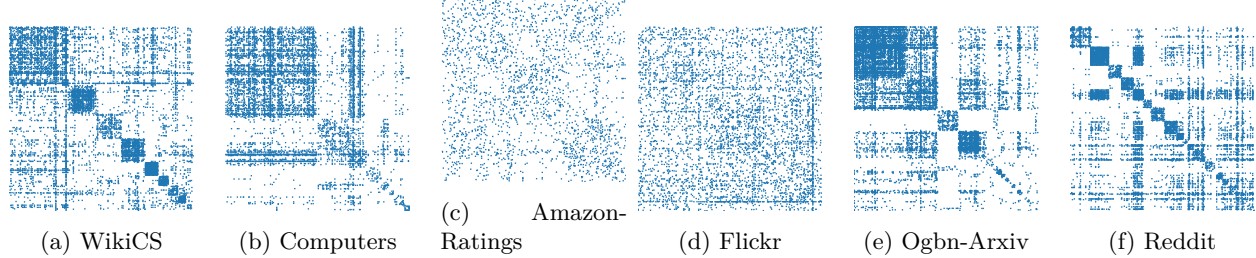

Figure 4: Visualization of Target graphs' adjacency matrices. The horizontal and vertical axes are node IDs ordered by class sizes. The majority classes are placed at the top-left corner.

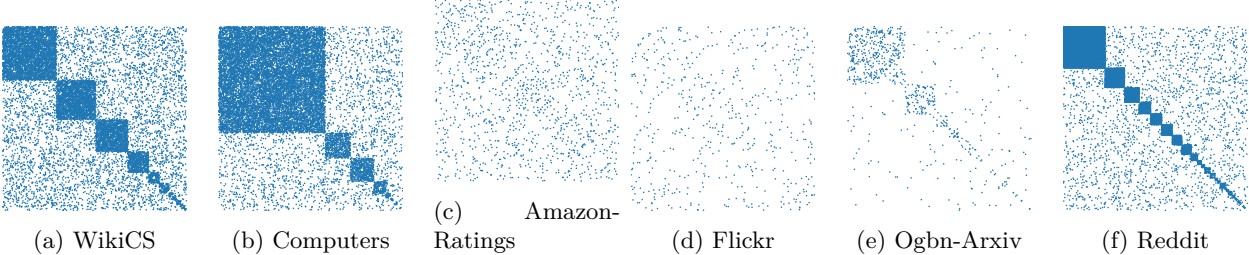

Figure 5: Synthetic adjacency matrices (SBM).

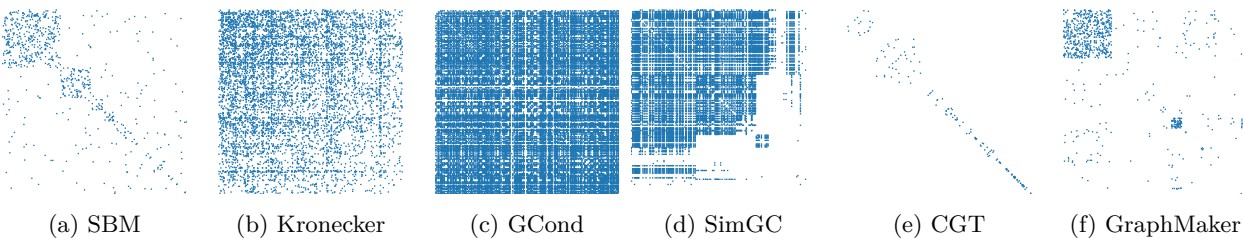

Figure 6: Ogbn-Arxiv generated by different methods.

**Existing methods can well capture the inter-class connectivity patterns.** We observe that existing methods, particularly the SBM model, can accurately capture the inter-class connectivity patterns in the original graphs, reflecting the different homophily patterns that exist in real-world graphs.

**Irrelevance between Structural Fidelity and Utility.** We notice that utility-focused models, such as GCond and SimGC, produce structures that significantly diverge from the original graph. In particular, they emphasize the connections within majority classes while ignoring many minority class connections. This inspires future endeavors in graph generation algorithms that focus on a particular aspect so that the synthetic structure can be better tuned to meet certain training objectives.

