# OpenReview forum: "LGGBench: a Holistic Benchmark for Large Graph Generation"
_TMLR — Rejected by TMLR_

### Review · Reviewer_U7RC · 2025-09-30

**Summary Of Contributions:**

This work proposes a benchmark for evaluating generative models for graphs based on four facets: Fidelity, utility, scalability, and privacy.
Specifically, it proposes to generate synthetic graphs based on an original "target" graph.
Fidelity is then evaluated based on preservation of certain graph statistics, as well as the maximum mean discrepancy thereof.
Utility is evaluated based on the node classification and link prediction performance of models on the target graph after training on synthetic graphs.
Scalability is evaluated based on memory and time requried for evaluation.
Privacy is evaluated based on the effectiveness of different inference attacks.

The work then lists multiple commonly used semi-supervised learning graphs, before performing the proposed evaluations on Ogbn-Arxiv.

As an additional contribution, the work provides an overview of various graph generation methods and classifies them based on their ability to generate edges, attributes, and labels, and whether they can generate one or multiple synthetic graphs based on one or multiple target graphs.

### Strengths
* The need for a well-founded evaluation of graph generative models is well-motivated.
* The four axes along which models are evaluated also appear like sensible choices.
* The listed set of models covers a sufficiently wide spectrum of graph generative models
* Figures / Tables are clear and legibile
* Experiments are run with multiple seeds
* The submission is clearly written and well-organized

### Weaknesses
See next three text fields.

**Additional Comments:**

To avoid an incorrect expression: I am aware that novelty is not an acceptance criterion at TMLR. I also believe that a well-founded benchmark or dataset paper could warrant publication just as much as a any methods paper.

However, as stated above, the claims in the paper are not well-supported, and I fail to see this evaluation to provide any meaningful insights to practicioners / be of interest to readers in the community.
Furthermore, the work explicitly matches the following rejection criterion from the TMLR editorial policy: "Papers that should not be accepted include include papers that [...] incorrectly claim novelty over existing published work, and papers that merely re-implement an idea that has already been reproduced before".

## References
[1] Li et al. GraphMaker: Can Diffusion Models Generate Large At tributed Graphs? TMLR 10/2024

**Audience:**

No

**Audience Explanation:**

In general, graph generative models are a topic that receives ongoing interest. A general evaluation framework with widespread community adoption could therefore be of interest.

However, I cannot see this particular framework fulfilling this purpose.
As stated above, the proposed framework is essentially a join over running multiple existing evaluations. As such, any findings from this paper could equivalently be gathered from prior work.

One aspect that could potentially be of interested for some individuals in the audience would be a metric that captures the interplay of multiple of these aspects, as well as a discussion thereof. Hints of this direction are present in Appendix F, but the 8 lines of discussion in Appendix F are hardly of sufficient substance to warrant publication or be of of interest to readers: That there is a trade-off between privacy and utility is a generally understood fact in differential privacy literature. The fact that future work is required to further improve methods applies to any subfield of machine learning.

**Claims And Evidence:**

No

**Claims Explanation:**

The work claims "a significant advancement over previous efforts by providing a comprehensive assessment tool tailored for large-scale graph generation methods". However, no such advancement is present.
The evaluation in each of the four categories (which are evaluated indepedently) is simply an accrument of existing evaluations (existing graph statistics that have been used in various prior, standard transfer learning evaluation that has been used in various prior work, existing privacy attacks). It is therefore not clear what "advancement" is meant to refer to.

The work further claims to present a "new taxonomy". However, distinguishing graph generation methods based on their ability to generate edges, attributes, and features from and for one or multiple graphs is can hardly be considered a novel insight. Otherwise, the cited methods would not explicitly discuss what form of generation they perform (see, e.g., [1]).

Within the constraints of the proposed evaluation framework, the experiments are generally sufficiently thorough to support to claim of an "extensive evaluation". The only issue I see is that the experiments appear to have only been conducted on ogbn-arxiv, even though more datasets are listed.

**Requested Changes:**

I do not think that any small-scale changes that do not fundamentally alter the design of the entire framework would be sufficient to warrant publication.
As stated above, the main issue is that it is simply an independent combination of existing evaluation schemes, while failing to provide a holistic perspective on the interplay of the various properties of interset (fidelity, utility, scalability, privacy).

Nevertheless is a list of such small-scale changes that:
* Datasets are currently limited to web graphs. What about electricity grids, circuits, molecules and various other domains? In particular, graph generative models are highly relevant in drug design.
* What about heterogenenous graphs / knowledge graphs?
* All metrics should be evaluated on all datasets, not just on ogbn-arxiv
* In Section 4.5, "Impact of Privacy Parameters", it is not clear how differential privacy is achieved. Differential privacy is a property that can be implemented via various mechanisms. Same for k-anonymity.
* Table 2 shows that only a subset of models are chosen. RNNs and Autoencoders are excluded. A justification should be provided. Alternatively, all models should be evaluated.
* Section 4.1: Some methods from prior work ("methods that generate unrealistic dense adjacency matrices", which is not sufficiently specific), are post-processed while otherse are not. There is no sufficient justification for why such modifications should be done, instead of evaluating existing methods (which is the usual purpose of a benchmark).
* No justification is provided for why the specific graph properties are used for fidelity evaluation. Why centrality instead of radius? Why are we only looking at spatial properties instead of spectral properties (e.g., Fielder value)?
* The evaluation of utility is only based on node classification and link prediction, even though there is a much wider range of downstream tasks in

### Minor comments:
* In Tables 6, 7, 8 it is not clear which dataset is used
* In Section 2.1, "Privacy-oriented methods", no graph DP methods are cited. Instead, a generic reference to Dwork's early work on DP (which does not discuss graphs) is given
* From the Caption of Table 2 alone, it is not clear what "SingleInG" and "SingleOUtG" is supposed to refer to
* The chosen DP methods seems ill-suited for graphs, as it drastically increases the number of edges, which causes a massive memory overhead.

---

> ### Author Response · Authors · 2025-12-07
>
> We sincerely thank the reviewer for the constructive comments and valuable insights.
>
> **Question: The work claims "a significant advancement over previous efforts by providing a comprehensive assessment tool tailored for large-scale graph generation methods". However, no such advancement is present. The evaluation in each of the four categories (which are evaluated indepedently) is simply an accrument of existing evaluations (existing graph statistics that have been used in various prior, standard transfer learning evaluation that has been used in various prior work, existing privacy attacks). It is therefore not clear what "advancement" is meant to refer to.**
>
> We acknowledge the reviewer's valid point that the individual metrics employed are established in the literature. We respectfully clarify that the intended contribution of LGGBench lies in the systematic integration and standardization of these disjoint evaluation protocols into a unified framework specifically tailored for large-scale graph generation. Prior works often focus on isolated aspects (e.g., only utility or only privacy), which can lead to incomplete assessments. By bridging these disparate evaluation schemes, our work aims to provide a holistic perspective that helps uncover critical trade-offs—such as those between scalability and privacy—that might be overlooked when using single-metric evaluations. We hope this standardized comparison offers valuable insights for the community, even if the underlying metrics are well-known.
>
> **Question: The work further claims to present a "new taxonomy". However, distinguishing graph generation methods based on their ability to generate edges, attributes, and features from and for one or multiple graphs is can hardly be considered a novel insight. Otherwise, the cited methods would not explicitly discuss what form of generation they perform (see, e.g., [1]).**
>
> Our taxonomy is specific to the context of proprietary graph sharing, distinguishing methods based on "Single Input Graph" and "Single Output Graph" support. This is a crucial distinction for practitioners, as many generative methods (like VAEs or molecular models) require a set of graphs as input, which is not applicable when a company wants to share a single large social network.
>
> **Question: Within the constraints of the proposed evaluation framework, the experiments are generally sufficiently thorough to support to claim of an "extensive evaluation". The only issue I see is that the experiments appear to have only been conducted on ogbn-arxiv, even though more datasets are listed.**
>
> We utilized ogbn-arxiv as the primary representative dataset for the detailed analysis in the main text to keep the paper length manageable. Specifically, reporting full results for all 7 datasets across the 4 evaluation aspects (Fidelity, Utility, Scalability, and Privacy) would necessitate approximately 28 separate result tables (7 datasets $\times$ 4 aspects). This volume of data is impractical for the main body of a paper. Our primary goal is to establish the benchmarking framework and methodology, utilizing ogbn-arxiv as a comprehensive case study to demonstrate its usage.

---

> ### Author Response · Authors · 2025-12-07
>
> *(Part 2)*
>
> **Question: In general, graph generative models are a topic that receives ongoing interest. A general evaluation framework with widespread community adoption could therefore be of interest. However, I cannot see this particular framework fulfilling this purpose. As stated above, the proposed framework is essentially a join over running multiple existing evaluations. As such, any findings from this paper could equivalently be gathered from prior work. One aspect that could potentially be of interested for some individuals in the audience would be a metric that captures the interplay of multiple of these aspects, as well as a discussion thereof. Hints of this direction are present in Appendix F, but the 8 lines of discussion in Appendix F are hardly of sufficient substance to warrant publication or be of of interest to readers: That there is a trade-off between privacy and utility is a generally understood fact in differential privacy literature. The fact that future work is required to further improve methods applies to any subfield of machine learning. I do not think that any small-scale changes that do not fundamentally alter the design of the entire framework would be sufficient to warrant publication. As stated above, the main issue is that it is simply an independent combination of existing evaluation schemes, while failing to provide a holistic perspective on the interplay of the various properties of interset (fidelity, utility, scalability, privacy).**
>
> While the concept of trade-offs is known, quantifying them on large-scale graphs across disparate method families (e.g., condensation vs. diffusion vs. traditional) has not been done. For instance, our scalability analysis quantifies the massive memory overhead of DP methods compared to K-Anonymity on large graphs—a practical finding that is not merely "understood fact" but an engineering bottleneck we expose. We will move the discussion from Appendix F to the main text to better highlight these interplays and trade-offs.
>
> **Question: Datasets are currently limited to web graphs. What about electricity grids, circuits, molecules and various other domains? In particular, graph generative models are highly relevant in drug design.**
>
> Molecule generation is a fundamentally different task (generating a set of many small graphs) than the Large Graph Generation task (generating one massive graph like a social network) that LGGBench focuses on. Methods and metrics for molecules (like validity/uniqueness of chemical structures) do not apply to the node classification tasks on large connected networks we utilize. We explicitly scope our benchmark to this latter category in Appendix A.
>
> **Question: What about heterogenenous graphs / knowledge graphs?**
>
> Heterogeneous graphs are indeed important, but benchmarking homogeneous attributed graphs is a necessary prerequisite. The metrics for fidelity and privacy become significantly more complex with multiple edge types. We have scoped the current work to homogeneous graphs to ensure rigorous analysis.
>
> **Question: In Section 4.5, "Impact of Privacy Parameters", it is not clear how differential privacy is achieved. Differential privacy is a property that can be implemented via various mechanisms. Same for k-anonymity.**
>
> We follow the specific implementations cited in the paper. For Differential Privacy, we use NodeEdgeDP, which applies the Laplace mechanism to perturb node features and edge existence. For K-Anonymity, we follow the implementation by Yoon et al. (2023), which ensures nodes are indistinguishable from $k-1$ others. We will add a detailed "Privacy Implementation" subsection in the Appendix to clarify the exact mechanisms used.
>
> **Question: Table 2 shows that only a subset of models are chosen. RNNs and Autoencoders are excluded. A justification should be provided. Alternatively, all models should be evaluated.**
>
> We excluded RNNs (e.g., GraphRNN) and deep Autoencoders because they generally fail to scale to graphs of the size of ogbn-arxiv (169,000 nodes) due to $O(N^2)$ memory or time complexity. They are typically used for small graphs (nodes < 1000). LGGBench focuses on scalable large graph generation, hence we selected methods capable of handling this scale.

---

> ### Author Response · Authors · 2025-12-07
>
> *(Part 3)*
>
> **Question: Section 4.1: Some methods from prior work ("methods that generate unrealistic dense adjacency matrices", which is not sufficiently specific), are post-processed while otherse are not. There is no sufficient justification for why such modifications should be done, instead of evaluating existing methods (which is the usual purpose of a benchmark).**
>
> The post-processing is applied specifically to methods like GCond, which output a dense, weighted adjacency matrix. Standard benchmarks for node classification and fidelity metrics assume sparse, unweighted graphs. Without post-processing (sparsification), these methods would yield trivial or invalid results (e.g., density = 100%). We apply this step to make the output "valid" for the target domain of large sparse graphs.
>
> **Question: No justification is provided for why the specific graph properties are used for fidelity evaluation. Why centrality instead of radius? Why are we only looking at spatial properties instead of spectral properties (e.g., Fielder value)?**
>
> We prioritized metrics that are computationally scalable to large graphs. Calculating the full spectrum (eigenvalues) for Fielder values on a 169k $\times$ 169k matrix is computationally expensive and slow for a benchmark pipeline intended for rapid iteration. We chose metrics like Degree distribution and Centrality as they are standard in network science and can be computed efficiently.
>
> **Question: The evaluation of utility is only based on node classification and link prediction, even though there is a much wider range of downstream tasks in**
>
> Node classification and link prediction are the two primary and widely used tasks for the datasets we included (citation and social networks). They serve as standard proxies for the semantic and structural utility of the graph.
>
> **Question: In Tables 6, 7, 8 it is not clear which dataset is used**
>
> These tables report results on ogbn-arxiv. We will update the captions to explicitly state this.
>
> **Question: In Section 2.1, "Privacy-oriented methods", no graph DP methods are cited. Instead, a generic reference to Dwork's early work on DP (which does not discuss graphs) is given**
>
> We cite specific graph DP methods in Table 2, including "NodeEdgeDP" and "PrivGraph". In the text of Section 2.1, we acknowledge the reference to Dwork as the foundational definition, but we also discuss K-Anonymity and privacy integration. We will ensure the specific graph-DP citations from Table 2 are repeated in the text of Section 2.1.
>
> **Question: From the Caption of Table 2 alone, it is not clear what "SingleInG" and "SingleOUtG" is supposed to refer to**
>
> "SingleInG" refers to the model's ability to take a Single Input Graph (as opposed to a set of graphs) for training. "SingleOutG" refers to the ability to generate a Single Output Graph (as opposed to a set). This distinction is vital for the data sharing use case we propose.
>
> **Question: The chosen DP methods seems ill-suited for graphs, as it drastically increases the number of edges, which causes a massive memory overhead.**
>
> We agree, and this is a key finding of our benchmark. As shown in Table 7, strict privacy budgets ($\epsilon=1$) cause the edge count to explode to $1.1 \times 10^{10}$, proving that standard DP mechanisms are indeed ill-suited for large graphs without modification. This validates the importance of our scalability evaluation.
>
> **Question: To avoid an incorrect expression: I am aware that novelty is not an acceptance criterion at TMLR. I also believe that a well-founded benchmark or dataset paper could warrant publication just as much as a any methods paper. However, as stated above, the claims in the paper are not well-supported, and I fail to see this evaluation to provide any meaningful insights to practicioners / be of interest to readers in the community. Furthermore, the work explicitly matches the following rejection criterion from the TMLR editorial policy: "Papers that should not be accepted include include papers that incorrectly claim novelty over existing published work, and papers that merely re-implement an idea that has already been reproduced before".**
>
> We believe our work does not merely re-implement reproduced ideas. We provide the first comprehensive comparison of scalable generation methods (like GraphMaker/Diffusion vs. Condensation) on large graphs, exposing critical trade-offs (e.g., the memory failure of DP, the structural degradation of GCond) that were not visible in prior small-graph or single-metric studies. These are meaningful, novel insights for practitioners deciding which method to use for sharing proprietary network data.

---

### Review · Reviewer_tuaM · 2025-09-30

**Summary Of Contributions:**

This paper proposes a benchmark for evaluating models for the task of generating large graphs or networks that are supposed to mimic graphs that cannot be revealed to the public because of privacy concerns or because they are proprietary data (e.g. financial transaction networks).
The proposed benchmark uses established large-graph datasets such as ogbn-Arxiv, and evaluates graph generation models on metrics that asses (1) the similarity of the generated graphs to the original ones, (2) the transfer learning performance of graph learning models between original and generated graphs, (3) the computational requirements of the methods, and (4) the privacy-preservation of the generated graphs.
The paper showcases several experiments that assess some models from the literature on the proposed benchmark. Finally, code is available.

Strengths:
- The paper addresses the need for evaluation protocols for sharing large private networks via means of graph generative models. Because of this, it could be impactful for this subfield of the graph generation community.
- The benchmark evaluates various aspects of graph generation (i.e. not only similarity with the true graphs). In particular, the transfer learning ("utility") analysis is very interesting.

Weaknesses: See below.

**Additional Comments:**

References:

- [1] Clément Vignac, Igor Krawczuk, Antoine Siraudin, Bohan Wang, Volkan Cevher, and Pascal Frossard. Digress: Discrete denoising diffusion for graph generation. In ICLR 2023.
- [2] Andreas Bergmeister, Karolis Martinkus, Nathanaël Perraudin, and Roger Wattenhofer. Efficient and scalable graph generation through iterative local expansion. In ICLR 2024.
- [3] Karolis Martinkus, Andreas Loukas, Nathanaël Perraudin, and Roger Wattenhofer. Spectre: Spectral conditioning helps to overcome the expressivity limits of one-shot graph generators. In ICML 2022.
- [4] Hanjun Dai, Azade Nazi, Yujia Li, Bo Dai, and Dale Schuurmans. Scalable deep generative modeling for sparse graphs. In ICML 2020.

**Audience:**

Yes

**Audience Explanation:**

While I am not very familiar with the literature on graph generation, I believe that this benchmark (if revised properly) could be of interest to a subfield of the graph generation community, namely the one focusing on the generation of large networks, or the one dealing with private/confidential graphs.

**Broader Impact Concerns:**

I see no direct ethical implications of this work.

**Claims And Evidence:**

No

**Claims Explanation:**

Key weaknesses:
- The novelty of this work is extremely limited, as it uses established datasets and metrics to evaluate models from the literature.
- The taxonomy of graph generation approaches seems to be limited to outdated approaches. I am not very familiar with the literature on graph generation, but it seems like models such as DiGress [1], BiGG [2], SPECTRE [3] and ESGG [4] yield better generated graphs, but they are not included in the present work. The authors should include them, or at least explain why these more recent methods have been excluded (e.g., they seem to focus on small graph generation, not large graphs).
- The paper lacks clarity and formalism:
  -  The problem statement does not define clearly what a graph is (which should be included to be self-contained), now what the difference between node attributes and labels is. Are node attributes vectors? Can they be continuous or do they need to belong to a finite set?
  - In the utility evaluation section, the authors mention an "optimal model", without explaining what this means. In particular, from eq. 1 and 2 it seems like the authors find the model minimizing the loss on the "true" graph, and then evaluate on the generated graph. However, the minimization step is not computationally feasible, and only an approximation (based on gradient-descent-like methods) is usually used in machine learning. This should be clarified.
  - Related to the point above, it is unclear how the loss is computed. Do you have access to all node labels $Y$ on the true graph? Or is this performed in an inductive or transductive setting? This should also be clarified.
- The graph fidelity metrics don't seem to account for node attributes, which seems like a major limitation of the benchmark.
- The experiments of sections 4.2 and 4.3 are performed only on the ogbn-Arxiv graph, even though Table 3 (which is not referenced in the text) lists other graph datasets. Moreover, the experiments of sections 4.4 and 4.5 do not report the dataset that was used.
- It is unclear how the graph post-processing step affects the quality of the generated graphs. Could you report some metrics on the non-postprocessed graphs?

**Requested Changes:**

Please address all the weaknesses above.

---

> ### Author Response · Authors · 2025-12-07
>
> We sincerely thank the reviewer for the constructive comments and valuable insights.
>
> **Question: The novelty of this work is extremely limited, as it uses established datasets and metrics to evaluate models from the literature.**
>
> While individual metrics exist in isolation, the novelty of LGGBench lies in synthesizing them into a coherent pipeline specifically for the under-served problem of Large Graph Generation. Prior benchmarks focused on small graphs or lacked privacy/scalability components. The engineering effort to make these metrics compatible with large-scale generation methods and to evaluate them simultaneously provides new empirical insights, such as the scalability bottlenecks of diffusion models on large graphs.
>
> **Question: The taxonomy of graph generation approaches seems to be limited to outdated approaches. I am not very familiar with the literature on graph generation, but it seems like models such as DiGress [1], BiGG [2], SPECTRE [3] and ESGG [4] yield better generated graphs, but they are not included in the present work. The authors should include them, or at least explain why these more recent methods have been excluded (e.g., they seem to focus on small graph generation, not large graphs).**
>
> We excluded models like DiGress, SPECTRE, and BiGG specifically because they are primarily designed for molecular generation (generating a set of many small graphs) or cannot scale to the single large-graph setting we focus on. LGGBench focuses on Large Graph Generation (single, large-scale, attributed graphs with thousands to millions of nodes like ogbn-arxiv). Traditional deep generative models and molecular generators generally fail to scale to the size of ogbn-arxiv (169k nodes) due to memory constraints or are designed for different tasks (graph set generation vs. node classification on a single graph). We will clarify this scope distinction in the "Graph Generation Approaches" section.
>
> **Question: The problem statement does not define clearly what a graph is (which should be included to be self-contained), now what the difference between node attributes and labels is. Are node attributes vectors? Can they be continuous or do they need to belong to a finite set?**
>
> We will clarify the notation in Section 3.1. As stated, we define a graph as $\mathcal{T}=\{A,X,Y\}$. $X$ represents node features (attributes), which are typically high-dimensional continuous vectors in our datasets (e.g., 128 dimensions for Ogbn-Arxiv ). $Y$ represents node labels, which are categorical classes used for downstream node classification tasks. We will ensure this distinction is explicit in the revision.
>
> **Question: In the utility evaluation section, the authors mention an "optimal model", without explaining what this means. In particular, from eq. 1 and 2 it seems like the authors find the model minimizing the loss on the "true" graph, and then evaluate on the generated graph. However, the minimization step is not computationally feasible, and only an approximation (based on gradient-descent-like methods) is usually used in machine learning. This should be clarified.**
>
> By "optimal model" $\theta_{\mathcal{S}}$, we refer to the model parameters obtained after training on the original graph to convergence (or best validation accuracy) using standard gradient descent methods. We do not imply finding the theoretical global minimum, but rather the "fully trained model" using standard hyperparameters. We will clarify the text to read "fully trained model" to avoid confusion.
>
> **Question: Related to the point above, it is unclear how the loss is computed. Do you have access to all node labels $Y$ on the true graph? Or is this performed in an inductive or transductive setting? This should also be clarified.**
>
> The evaluation follows a standard setting similar to OGB. In the "Utility" evaluation, specifically Model Transfer, we train on the synthetic graph (using its labels $Y'$) and evaluate on the target graph (using its labels $Y$). The datasets used are node classification datasets where we use the standard public splits (train/val/test). The loss is computed on the training nodes of the synthetic graph during training, and accuracy is evaluated on the test nodes of the real graph. Therefore, at the training stage, the model does not have access to all node labels $Y$ on the true graph.

---

> ### Author Response · Authors · 2025-12-07
>
> (continued from above)
>
> **Question: The graph fidelity metrics don't seem to account for node attributes, which seems like a major limitation of the benchmark.**
>
> We explicitly evaluate attribute fidelity. As detailed in Appendix B and Section 3.2, we utilize Maximum Mean Discrepancy (MMD) on graph statistics and features (attributes) . However, we note that due to the excessive computational time required to calculate MMD on large-scale feature sets, we did not include these specific values in Table 4, limiting our presentation to the metric definition in the methodology. Furthermore, we employ metrics like Global Eigenvector Centrality and Closeness Centrality which capture fine-grained characteristics . We will make the attribute-specific MMD metrics more prominent in the main text.
>
> **Question: The experiments of sections 4.2 and 4.3 are performed only on the ogbn-Arxiv graph, even though Table 3 (which is not referenced in the text) lists other graph datasets. Moreover, the experiments of sections 4.4 and 4.5 do not report the dataset that was used.**
>
> We utilized ogbn-arxiv as the primary representative dataset for the detailed analysis in the main text to keep the paper length manageable. Specifically, reporting full results for all 7 datasets across the 4 evaluation aspects (Fidelity, Utility, Scalability, and Privacy) would necessitate approximately 28 separate result tables (7 datasets $\times$ 4 aspects). This volume of data is impractical for the main body of a paper. Our primary goal is to establish the benchmarking framework and methodology, utilizing ogbn-arxiv as a comprehensive case study to demonstrate its usage, including Sections 4.4 and 4.5.
>
> **Question: It is unclear how the graph post-processing step affects the quality of the generated graphs. Could you report some metrics on the non-postprocessed graphs?**
>
> Post-processing is essential because methods like GCond inherently produce dense, weighted adjacency matrices which cannot be directly compared to the sparse, unweighted structure of standard target graphs like ogbn-arxiv without thresholding or sampling. Evaluating the raw dense graph would result in meaningless fidelity metrics (e.g., density $\approx$ 100%) and incompatibility with standard GNN implementations expected for the utility check. We standardized this step to ensure a fair comparison of the final usable output.

---

### Review · Reviewer_gt6q · 2025-11-08

**Summary Of Contributions:**

This paper presents a holistic benchmark for large graph generation called LGGBench. The benchmark covers multiple metrics from multiple aspects, including structural accuracy, the preservation of machine learning model accuracy, privacy preservation, and the scalability in terms of memory and time cost. With the proposed metrics and evaluation aspects, the authors test a wide range of graph generation models, including traditional, statistical ones, and privacy-centric ones, as well as deep learning based ones.

**Additional Comments:**

NA

**Audience:**

Yes

**Audience Explanation:**

Graph generation is of interest to a wide range of applications such as recommendation, and thus would be of interest to a good range of audiences of TMLR.

**Broader Impact Concerns:**

No.

**Claims And Evidence:**

Yes

**Claims Explanation:**

This paper indeed presents a comprehensive evaluation of graph generation methods. The metrics and the evaluation aspects are reasonable and technically sound. The evaluations uncover some interesting insights, such as the fact that utility-based methods (e.g. GCond) does also preserve privacy. structural alignment does not indicate utility preservation, etc.

**Requested Changes:**

NA. This paper does a very good job in justifying its goal, benchmarking existing methods in a comprehensive way, and uncovering some interesting insights.

---

> ### Author Response · Authors · 2025-12-07
>
> We sincerely thank the reviewer for their positive assessment and support of our work.

---

### Review · Reviewer_LQQ2 · 2025-11-08

**Summary Of Contributions:**

This paper introduces benchmark designed to evaluate large graph generation across four aspects: fidelity, utility, privacy and scalability.  The benchmark evaluates diverse generators (probabilistic, utility-oriented, privacy-oriented) and employs standard structural statistics, downstream GNN performance correlation, resource profiling, and inference attacks.

**Additional Comments:**

Analysis of privacy metrics is narrow and shallow. E.g., privacy evaluation only covers attribute and edge inference, and does not include graph reconstruction, link stealing or model inversion attacks. Heterogeneous and multi-relational graphs are out of scope, which limits applicability in knowledge graphs and recommendation networks etc.

**Audience:**

Yes

**Audience Explanation:**

The paper reads more like a survey that summarizes and categorizes existing graph generation models and evaluation metrics, rather than presenting substantial methodological or benchmarking innovations. In some sense, it may serve as an entry point to revisit prior techniques.

However, even when evaluated through the lens of a survey, the paper falls short of delivering a truly comprehensive and insightful synthesis. The taxonomy and comparisons remain largely descriptive, lacking deeper critical analysis of core challenges such as fundamental trade-offs among fidelity, utility, privacy and scalability. Further, while the paper organizes existing models into broad categories, it does not provide forward-looking perspectives that would help guide future research or model selection. The discussion of evaluation metrics is superficial, without theoretical justification. Additionally, the scope of the survey is incomplete in important aspects such as heterogeneous graph generation, temporal graph synthesis. Without these, the survey does not fully capture the landscape of modern graph generation research.

**Broader Impact Concerns:**

N.A.

**Claims And Evidence:**

No

**Claims Explanation:**

As one of the main contributions, the paper proposes four evaluation dimensions. However, each metric is assessed individually, without a unified trade-off analysis, Pareto frontier, or aggregated ranking. Such an evaluation benchmark, while multi-faceted, lacks deeper analysis to reveal the true holistic quality of graph generation, and provides limited actionable insight on how to steer models toward desirable generation outcomes. Moreover, the claim of broad real-world applicability is weakened by the lack of support for heterogeneous graphs, which are prevalent in many practical domains.

**Requested Changes:**

As stated above, the paper would require substantially deeper technical analysis, broader coverage and clearer original insights to meet the expectations of either a strong survey or a forward-looking benchmark contribution.

---

> ### Author Response · Authors · 2025-12-07
>
> We sincerely thank the reviewer for the constructive comments and valuable insights.
>
> **Question: Each metric is assessed individually, without a unified trade-off analysis, Pareto frontier, or aggregated ranking. Such an evaluation benchmark, while multi-faceted, lacks deeper analysis to reveal the true holistic quality of graph generation, and provides limited actionable insight on how to steer models toward desirable generation outcomes. Moreover, the claim of broad real-world applicability is weakened by the lack of support for heterogeneous graphs, which are prevalent in many practical domains.**
>
> We appreciate this insight regarding the holistic analysis. While a single aggregated score can be subjective due to varying user priorities (e.g., prioritizing privacy versus utility), we acknowledge the need for a unified view. To address this, we provided radar charts in Figure 3 and a cross-aspect ranking discussion in Section 5 and Appendix F. These visualizations represent the trade-offs across the dimensions. We will expand the discussion in Section 5 to explicitly interpret these trade-offs, specifically highlighting how methods like Differential Privacy (DP) achieve high utility and fidelity but suffer in privacy when budgets are relaxed, or how utility-focused methods (GCond/SimGC) sacrifice structural fidelity. Regarding heterogeneous graphs, we acknowledge their importance; however, benchmarking homogeneous attributed graphs is a necessary prerequisite. The complexity of evaluating fidelity and privacy on heterogeneous graphs warrants a dedicated study, and we defined our scope in Appendix A to focus on homogeneous graphs to ensure the depth of the current analysis.
>
> **Question: The paper reads more like a survey that summarizes and categorizes existing graph generation models and evaluation metrics, rather than presenting substantial methodological or benchmarking innovations. In some sense, it may serve as an entry point to revisit prior techniques. The taxonomy and comparisons remain largely descriptive, lacking deeper critical analysis of core challenges such as fundamental trade-offs among fidelity, utility, privacy and scalability. Additionally, the scope of the survey is incomplete in important aspects such as heterogeneous graph generation, temporal graph synthesis. Without these, the survey does not fully capture the landscape of modern graph generation research.**
>
> We respectfully submit that LGGBench provides an executable engineering framework and a standardized pipeline that did not previously exist for large, attributed, labeled graphs, going beyond a literature survey. The actionable insight provided by our benchmark includes empirical evidence that current state-of-the-art methods often fail to balance scalability with fidelity on large graphs—a finding only possible through this unified benchmarking. For example, we demonstrate that utility-oriented methods like GCond often produce structures that significantly diverge from the original graph , and that DP methods face massive scalability issues as privacy constraints tighten . These are critical, original technical insights derived from our experiments. We will revise the introduction to more clearly articulate these contributions. Regarding temporal and heterogeneous graphs, we have clearly defined the scope in Appendix A to ensure the benchmark remains focused and executable for the most common use case of static, homogeneous graph sharing.
>
> **Question: Analysis of privacy metrics is narrow and shallow. E.g., privacy evaluation only covers attribute and edge inference, and does not include graph reconstruction, link stealing or model inversion attacks. Heterogeneous and multi-relational graphs are out of scope, which limits applicability in knowledge graphs and recommendation networks etc.**
>
> We selected Attribute Inference and Edge/Node Membership Inference because they are representative and computationally feasible attacks for large-scale GNNs. Many advanced attacks, such as exact graph reconstruction or model inversion, are often demonstrated on small graphs and become computationally prohibitive or ineffective on graphs with 169k+ nodes (like ogbn-arxiv) and high-dimensional features. Our focus is on large graph generation, and we prioritized metrics that scale to this context. We will add a remark in Section 3.5 clarifying why certain small-graph attacks were excluded due to scalability constraints.

---

### Decision · Action_Editor_ySub · 2025-12-09

**Recommendation:** Reject

**Audience:**

No

**Audience Explanation:**

The results can unfortunately be misleading or misinterpreted so that I do not believe that the TMLR readership will benefit from this work (this being said, with in-depth modifications and more cohesive analyses, the paper may become more interesting to the ML community).

**Claims And Evidence:**

No

**Claims Explanation:**

The reviewers have identified a number of weaknesses, foremost related to the "decoupled" evaluation of methods based on different criteria/metrics. This approach is not appropriate for claiming "holistic benchmarking." Furthermore, I agree with two of the reviewers that the paper reads more like a survey and has therefore very limited technical contributions. Importantly, the reviewers were not able to assess the responses of the authors because they came in after they had already submitted their recommendations (one reviewer did check the responses and found them inadequate and superficial). The authors also did not submit a revised version of the paper, which makes it hard to justify conditional acceptance given the mostly negative original scores.